# Unraveling the mechanisms behind the enhanced efficacy of β-lactam-based sideromycins
Evelyne Lacasse [1], Renaud Binette[2], Héloïse Guibout[1], Riu Liu[3,4], Yun-Ming Lin[3], Manuka Ghosh[3], Pierre-Luc Boudreault [5], Marvin J. Miller [3,4] ✉ & François Malouin [1] ✉

Previous studies have explored combining β-lactams with siderophores to create Trojan horse molecules that can penetrate the outer membrane of Gram-negative bacteria via TonB-dependent transporter (TBDT). While the main advantage explaining their enhanced antibiotic activity is believed to be improved membrane permeability, other factors remain underexplored. This study evaluates three siderophore-β-lactam compounds: a bis-catechol siderophore linked to ampicillin or loracarbef, and a mixed bis-catechol-mono-hydroxamate siderophore linked to cefaclor. Minimal inhibitory concentrations showed that siderophore conjugation could enhance β-lactam efficacy by over 8000-fold. Comparison with unconjugated β-lactams revealed a complex interplay between β-lactamase susceptibility, competition with endogenous siderophore, membrane uptake, and binding to penicillin-binding proteins (PBPs). Enhanced PBP binding, particularly in *Escherichia coli*, emerged as a key factor contributing to improved bacterial inhibition by siderophore-β-lactam conjugates. Overall, the study provides insights into how siderophore conjugation enhances β-lactam activity and the therapeutic potential of the conjugates as narrow or broad-spectrum antibiotics.

Since the discovery of penicillin, β-lactam antibiotics have remained critically important to human health. To enhance their efficacy and pharmacology, and to counteract problematic resistance mechanisms, β-lactams have undergone numerous synthetic modifications[1]. Unfortunately, bacteria have concurrently adapted and developed resistance mechanisms against these newly developed β-lactams. Today, the World Health Organization (WHO) AWaRe antibiotic classification has reserved three β-lactams among the eight antibiotics for last-line use: ceftazidime/avibactam, meropenem/vaborbactam, and cefiderocol[2]. Respectively, they are a third-generation cephalosporin and a carbapenem, both combined with a β-lactamase inhibitor, and a synthetic β-lactam-based sideromycin. Sideromycins, exemplified by the natural antibiotic albomycin[3] and the synthetic cefiderocol (formerly S-649266)[4], combine a siderophore moiety with an antibiotic (Fig. 1a, b). Bacterial siderophores are a class of diverse molecules exhibiting high affinity for iron ($Fe^{3+}$) produced and excreted by microorganisms[5]. Natural and synthetic sideromycins mimic siderophores produced by bacteria and exploit their iron transporters as cellular entry points[6]. Since bacteria require iron to colonize an environment or a host, hijacking their iron

uptake systems for antibiotic delivery offers a promising means of combating them. Moreover, sideromycins represent a way to specifically permeate the Gram-negative outer membrane, which is a natural barrier against many antibiotic classes.

In Gram-negative bacteria, siderophore uptake is facilitated by TonB-dependent transporter (TBDT). TBDT requires a TonB-ExbB-ExbD protein complex and the proton motive force to transport siderophores into the periplasm. TBDTs are redundant across species and differ in their specificity for siderophore ligands[7]. For instance, the TBDT FepA from *Escherichia coli* has a high and specific affinity for its endogenous siderophore enterobactin[8], while CirA can also bind cobalamin and catechol structures like the enterobactin biosynthesis intermediate 2,3-dihydroxybenzoic acid (2,3-DHBA)[9,10]. This broad ligand recognition constitutes a valuable feature for the design of antibiotic-siderophore conjugates, i.e., sideromycins. As the entire siderophore is not always needed for TBDTs recognition, this enables the design of sideromycins with a common siderophore moiety recognized by multiple TBDTs. Conversely, mimicking a particular siderophore with affinity toward a specific bacterial TBDT could enable the synthesis of narrow-spectrum antibiotics.

[1]Département de biologie, Faculté des Sciences, Université de Sherbrooke, Sherbrooke, QC, Canada. [2]Département de chimie, Faculté des Sciences, Université de Sherbrooke, Sherbrooke, QC, Canada. [3]Hsiri Therapeutics, Media, PA, USA. [4]Department of Chemistry and Biochemistry, University of Notre Dame, Notre Dame, IN, USA. [5]Département de Pharmacologie-Physiologie, Institut de Pharmacologie de Sherbrooke, Faculté de Médecine et des Sciences de la santé, Université de Sherbrooke, Sherbrooke, QC, Canada. ✉e-mail: mmiller1@nd.edu; francois.malouin@usherbrooke.ca

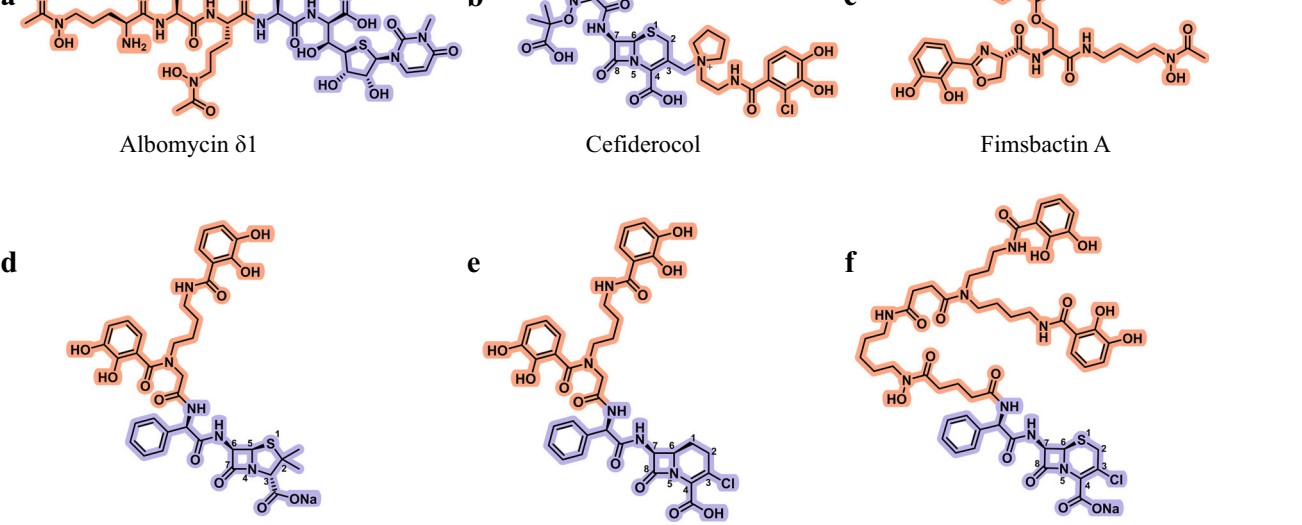

**Fig. 1 | Structure of conjugated β-lactams and molecules cited in this study.** Represented in orange are the siderophores and in purple the antibiotic components; **a** the natural antibiotic albomycin δ1 conjugates a trihydroxamate (ferrichrome-like) to a seryl tRNA inhibitor, **b** the synthetic antibiotic cefiderocol combines a mono-chloro-catechol to a ceftazidime/cefepime β-lactam moiety, **c** the siderophore fimsbactin A is an example of natural bis-catechol, **d** the bis-catechol (azotochelin)-ampicillin (BAMP), **e** the bis-catechol (azotochelin)-loracarbef (BLOR), and **f** the mixed ligand bis-catechol-mono-hydroxamate-cefaclor (MCEF), are also shown.

A low iron concentration triggers expression of TBDTs and siderophore uptake by bacteria. In aqueous solution and at neutral pH, iron solubility is low and ferric iron concentration is estimated to be around $10^{-18}$ M[11]. This concentration is far below the iron concentration needed to support bacterial growth, estimated to be around $10^{-6}$ M[11,12]. Iron availability for microbial pathogens is even lower in mammals due to active scavenging, sequestration, and storage of free iron by the host[13]. Antibiotic-siderophore conjugates are thus expected to show improved inhibitory activities in an iron-restricted mammalian environment[14]. Moreover, TBDTs constitute larger pores than nonselective porins, which are the principal entry points for β-lactams in many bacteria[15]. Studies have demonstrated that siderophore-antibiotic conjugates exceeding the 600 Da cutoff of porins can cross the Gram-negative outer membrane via TBDTs and accumulate in the periplasm[16,17]. Accordingly, β-lactam-siderophore conjugates, even with higher molecular weights than their unconjugated counterparts, are expected to pass through the appropriate TBDTs and access the periplasm. Once there, these β-lactam-based sideromycins can target penicillin-binding proteins (PBPs), bind to the transpeptidation domain of their periplasmic active site and thereby inhibiting peptidoglycan synthesis. PBPs are traditionally categorized by molecular weight, with high molecular weight (HMW) PBPs (e.g., PBP1, 1a, 1b, 2, 3) being generally implicated in vital peptidoglycan assembly, while low molecular weight (LMW) PBPs mostly have peptidoglycan maintenance and rearrangement functions[18].

Over the years, reports have shown that conjugating synthetic siderophores with β-lactams preserves and, sometimes, enhances their antimicrobial activity[19]. The recently FDA-approved cefiderocol, which combines a ceftazidime/cefepime β-lactam moiety with a mono-chloro-catechol siderophore mimetic, is particularly effective against resistant bacteria producing carbapenemases, extended spectrum β-lactamases (ESBL) and metallo-β-lactamases[20,21]. It has also been demonstrated that cefiderocol uptake in *Pseudomonas aeruginosa* occurs via TBDTs[21], highlighting the potential of this transport mechanism.

To investigate the mechanisms underlying the enhanced potency conferred by adding a siderophore to an antibiotic, this study examined three siderophore-β-lactam (SID-βL) conjugates: a bis-catechol siderophore, azotochelin[22], conjugated to ampicillin (BAMP)[23] or conjugated to loracarbef (BLOR), and a mixed ligand bis-catechol-mono-hydroxamate siderophore[24] conjugated to cefaclor (MCEF); see Fig. 1d–f. The design of the mixed siderophore used in MCEF was inspired by fimsbactin (Fig. 1c), a siderophore synthesized by *Acinetobacter baumannii*[24]. The synthetic siderophores used in this study were the result of many years of optimization. Previous studies have demonstrated that their conjugation to β-lactam, increased the antibiotic potency, against clinically relevant bacterial species, by over 500-fold compared to the unconjugated one[19]. In contrast, other siderophores, such as those based on hydroxamates, exhibit only low to moderate activity when conjugated to β-lactam, and the potency of these conjugates does not surpass that of the unconjugated β-lactam[25].

In this study, comparative analyses were done to compare both unconjugated β-lactams and SID-βL conjugates in relation to (1) β-lactamases, (2) efflux pumps, (3) their net uptake through the outer membrane, and (4) their affinity for PBPs. These four aspects were examined across four different Gram-negative species classified as WHO priority 1 for the development of new antibiotics: *Escherichia coli*, *Klebsiella pneumoniae*, *Pseudomonas aeruginosa*, and *Acinetobacter baumannii*[26]. Against the *Enterobacterales*, exemplified by *E. coli* and *K. pneumoniae*, the increased binding to PBPs was the major factor involved in the increased activity of conjugates. However, only increased uptake was necessary to observe an enhanced activity against *P. aeruginosa* and *A. baumannii*. These insights are expected to guide the rational design of future molecules.

## Results

### Iron concentration has a greater impact on the activity of SID-βL conjugates against *P. aeruginosa* and *A. baumannii* than against *E. coli* and *K. pneumoniae*

The potency of SID-βL conjugates against different bacterial species (described in Table S1) was first evaluated in MHBCA and in iron-depleted (ID-) MHBCA. To ensure that iron was an element limiting growth in ID-MHBCA, bacterial growth was monitored over 24 h by measuring optical density at 600 nm with increasing concentrations of supplemental iron to complement the depletion of ID-MHBCA (Fig. S1). The addition of iron to a $Fe^{3+}$ (from $FeCl_3$) concentration of 0.1 and 1 µg/mL ($1.8 \times 10^{-6}$ and

**Table 1 | Minimal inhibitory concentration (MIC) of unconjugated and conjugated β-lactams against Gram-negative bacteria**

| Antibiotic[b] and MIC ratio | MIC[a] determined in MHBCA (+) and ID-MHBCA (−) | | | | | | | |
|---|---|---|---|---|---|---|---|---|
| | *E. coli* | | *K. pneumoniae* | | *P. aeruginosa* | | *A. baumannii* | |
| | ATCC 25922 | | ATCC 13883 | | ATCC 27853 | | ATCC 19606 | |
| | + | - | + | - | + | - | + | - |
| AMP | 13 | 13 | >200 | >200 | >200 | >200 | >200 | >200 |
| BAMP | 0.1 | 0.05 | >200 | >200 | 1.6 | 0.1 | 0.4 | 0.2 |
| MIC ratio: AMP/BAMP | 128 | 256 | ND[c] | ND | >128 | >2048 | >512 | >1024 |
| LOR | 3.1 | 1.6 | 1.6 | 1.6 | >200 | >200 | >200 | >200 |
| BLOR | 0.024 | 0.012 | 0.2 | 0.1 | 13 | >200 | 0.05 | 0.024 |
| MIC ratio: LOR/BLOR | 128 | 128 | 8 | 16 | >16 | ND | >4096 | >8192 |
| CEF | 3.1 | 3.1 | 3.1 | 3.1 | >200 | >200 | >200 | >200 |
| MCEF | [6.3-25][d] | >200 | >200 | >200 | >200 | >200 | 1.6 | 0.2 |
| MIC ratio: CEF/MCEF | ND | <0.016 | <0.016 | <0.016 | ND | ND | >128 | >1024 |

[a]MICs are given in μM, +; MHBCA, −; iron-deprived (ID)-MHBCA.
[b]Antibiotics: AMP ampicillin, BAMP bis-catechol-ampicillin, CEF cefaclor, MCEF mixed bis-catechol-mono-hydroxamate-cefaclor, LOR loracarbef, BLOR bis-catechol-loracarbef.
[c]ND: not determined.
[d]The MIC range enclosed in brackets means that growth was observed at lower and higher concentrations (Eagle effect).

$1.8 \times 10^{-5}$ M) was necessary to restore the growth of all bacteria, except *K. pneumoniae* which showed no statistical difference in any tested condition.

For each bacterial species, the MIC of at least one of the tested β-lactams was greatly improved when conjugated to a siderophore moiety, as revealed by the fold-improvement ratios from 8 to >8192 reported for wild-type strains in Table 1. It was expected that a lower iron concentration would increase the activity of the SID-βL conjugates due to a higher expression of TBDTs and enhanced outer membrane uptake. However, Table 1 shows only two cases where a low iron concentration enhanced the SID-βL activity by more than fourfold when compared to the non-depleted medium. In the first case, BAMP showed a lower MIC against *P. aeruginosa*, in ID-MHBCA (MIC of 0.1 μM) compared to that measured in MHBCA (MIC of 1.6 μM). In the second case, the low iron concentration also improved the activity of MCEF in ID-MHBCA (MIC 0.2 μM) compared to that seen in MHBCA (1.6 μM) against *A. baumannii*. However, a dose-dependent increase in the MICs of the conjugates was observed in all species following iron supplementation of chelex treated ID-MHBCA medium (0, 0.1, and 1 μg/mL) (see Table S2). Supplementation with 0.1 μg/mL ($1.8 \times 10^{-6}$ M) resulted in MIC values equivalent to those measured in untreated MHBCA. This correlates with the previously measured iron concentration of 0.24 μg/mL ($4.3 \times 10^{-6}$ M) in this medium[21]. Moreover, supplementation with 1 μg/mL ($1.8 \times 10^{-5}$ M) led to a further increase in MICs, correlating with a decrease TBDTs expression compared to MHBCA.

## BAMP and BLOR have broad-spectrum activity, while MCEF has selective activity against *A. baumannii*

BAMP was active against all species except *K. pneumoniae*, which was also resistant to the unconjugated AMP. It was the only conjugate able to display a MIC against *P. aeruginosa* in ID-MHBCA (Table 1). BLOR was the only compound that induce a measurable MIC in all the species tested in MHBCA. BLOR was the conjugate with the highest gain in activity against *A. baumannii* (up to 8192-fold in ID-MHBCA). It was also the only conjugate active against *K. pneumoniae*. Although *K. pneumoniae* was susceptible to unconjugated LOR (MIC of 1.6 μM), the addition of the bis-catechol moiety to LOR improved its MIC to 0.2 μM. On the contrary, Table 1 shows that CEF alone is active against *E. coli* and *K. pneumoniae*, but its conjugation to the mixed siderophore importantly reduced its activity against these species, perhaps because of siderophore selectivity. Nevertheless, MCEF was still partly active against *E. coli* in MHBCA (growth inhibition in the concentration range indicated in brackets in Table 1) although growth was observed at higher concentrations (>25 μM). It is suspected that this phenomenon is the already documented Eagle effect. In *Proteus vulgaris*, this effect as been attributed to the induction of the β-lactamase[27]. Whereas, in *E. coli*, this phenomenon as been attributed to the SOS response and the upregulation of gene implicated in enterobactin synthesis and import[28]. In ID-MHBCA, MCEF completely lost its activity against *E. coli*. This inactivity was further investigated by the study of relevant mutants in the next section.

## Functional TBDTs are mandatory for the uptake of SID-βL conjugates in *E. coli*

To verify the importance of TBDTs for the activity of conjugates against *E. coli*, mutants from the Keio collection were evaluated (see description in Table S1)[29]. MICs for each mutant were compared to the MICs for the parental *E. coli* strain BW25113 (Table 2). The deletion of *tonB* was the most impactful for the activity of conjugates. It induced resistance against BAMP and BLOR, with a MIC of 50 μM compared to 0.05 and 0.012 μM, respectively, against *E. coli* BW25113 in MHBCA. Meanwhile, the MIC against MCEF only increased from 13 to 25 μM. The *tonB* deletion caused a major growth defect in ID-MHBCA, preventing MIC measurements in this iron-depleted medium. Deletion of specific iron transporter gene (*fepA*, *fiu*, *fhuA*, or *fecA*) had a mild effect on the activity of the unconjugated β-lactams, with two- and four-fold variation from the wild-type strain. The most important TBDT for the activity of all SID-βL conjugates was CirA in both MHBCA and ID-MHBCA media. The deletion of *cirA* induced MICs of 13 μM for BAMP, 13 μM for BLOR, and >200 μM for MCEF in MHBCA. The deletion of *fiu*, *fepA* or *fhuA* raised MICs of BLOR and BAMP between 0.1 and 0.4 μM in MHBCA, and only the deletion of *fiu* had an impact on MCEF, raising its MIC to 100 μM in MHBCA. In contrast, deletion of *fecA* increased the activity of BAMP and BLOR, leading to MICs of ≤0.003 μM in MHBCA. However, the MIC of MCEF for this mutant remained unchanged compared to BW25113 (13 μM). Notably, despite the clear role of CirA and Fiu for the activity of MCEF in MHBCA, no single TBDT deletion could explain the total loss of MCEF activity in ID-MHBCA. This led to the hypothesis of a direct competition between MCEF and enterobactin, a siderophore naturally synthesized by *E. coli* in iron-depleted environments.

## Altering enterobactin biosynthesis increases the activity of MCEF

As shown in Table 2, *E. coli* BW25113 loses its susceptibility to the MCEF conjugate in ID-MHBCA (MIC >200 μM). We thus investigated the possibility that the absorption of naturally produced enterobactin by *E. coli* grown in ID-MHBCA would compete with MCEF and reduce its activity. For this purpose, we used the *E. coli* ΔentE mutant strain that cannot synthesize enterobactin. Table 2 shows that *E. coli* ΔentE was indeed the only mutant susceptible to MCEF in ID-MHBCA with a MIC of 1.6 μM. This means that the resistance to MCEF in ID-MHBCA is, at least, partially due

**Table 2 | Implication of different TBDTs, enterobactin, and the AmpC β-lactamase in the activity of unconjugated and conjugated β-lactams against _E. coli_ strains**

| Strain | MIC[a] of parental _E. coli_ (BW25113) and mutant strains as determined in MHBCA (+) or ID-MHBCA (−) medium | | | | | | | | | | | |
| --- | --- | --- | --- | --- | --- | --- | --- | --- | --- | --- | --- | --- |
| | AMP | | BAMP | | LOR | | BLOR | | CEF | | MCEF | |
| | + | − | + | − | + | − | + | − | + | − | + | − |
| BW25113 | 13 | 13 | 0.05 | 0.024 | 6.3 | 6.3 | 0.012 | 0.006 | 6.3 | 6.3 | 13 | >200 |
| ΔtonB | 13 | ND[b] | **50**[c] | ND | 6.3 | ND | **50** | ND | 13 | ND | 25 | ND |
| ΔcirA | 13 | 25 | **13** | **[0.2–13]**[d] | 13 | 13 | **13** | **0.2** | 13 | 25 | **>200** | >200 |
| ΔfepA | 13 | 13 | 0.2 | 0.024 | 6.3 | 13 | **0.1** | 0.012 | 13 | 25 | 13 | >200 |
| Δfiu | 13 | 25 | 0.2 | 0.024 | 6.3 | 6.3 | **0.1** | 0.024 | 13 | 13 | **100** | >200 |
| ΔfhuA | 13 | 25 | **0.4** | **0.1** | 6.3 | 6.3 | **0.2** | 0.024 | 13 | 13 | 13 | >200 |
| ΔfecA | 13 | 13 | **≤0.003** | 0.006 | 3.1 | 6.3 | **≤0.003** | **≤0.003** | 6.3 | 6.3 | 13 | >200 |
| ΔentE | 13 | 13 | 0.2 | 0.05 | 6.3 | 6.3 | 0.05 | 0.012 | 6.3 | 6.3 | 13 | **1.6** |
| ΔampC | 6.3 | 6.3 | 0.024 | 0.012 | 3.1 | 3.1 | 0.006 | 0.006 | 3.1 | 3.1 | 13 | >200 |

[a]MICs are given in µM, +; MHBCA, −; iron-deprived (ID)-MHBCA, AMP ampicillin, BAMP bis-catechol-ampicillin, CEF cefaclor, MCEF mixed bis-catechol-mono-hydroxamate-cefaclor, LOR loracarbef, BLOR bis-catechol-loracarbef.
[b]ND: not determined.
[c]Data in bold highlight a > fourfold difference between mutant and parent strains in the same assay conditions.
[d]The MIC range enclosed in brackets means that growth was observed at lower and higher concentrations (Eagle effect).

## Table 3 | Impact of β-lactamase inactivation on the MICs of unconjugated and conjugated β-lactams

| Antibiotic[c] | MICs[a] with or without a β-lactamase inhibitor (βI)[b] determined in MHBCA (+) and ID-MHBCA (−) media | | | | | |
| --- | --- | --- | --- | --- | --- | --- |
| | _K. pneumoniae_ ATCC 13883 | | _P. aeruginosa_ ATCC 27853 | | _A. baumannii_ ATCC 19606 | |
| | + | − | + | − | + | − |
| AMP | >200 | >200 | >200 | >200 | >200 | >200 |
| AMP + βI | **6.3**[d] | **6.3** | **100** | **100** | >200 | >200 |
| BAMP | >200 | >200 | 1.6 | 0.1 | 0.4 | 0.2 |
| BAMP + βI | **1.6** | **0.4** | 1.6 | 0.1 | 0.2 | 0.1 |
| LOR | 1.6 | 1.6 | >200 | >200 | >200 | >200 |
| LOR + βI | 1.6 | 1.6 | >200 | >200 | >200 | >200 |
| BLOR | 0.2 | 0.1 | 13 | >200 | 0.05 | 0.024 |
| BLOR + βI | 0.2 | 0.1 | 6.3 | **0.8** | 0.012 | 0.012 |
| CEF | 3.1 | 3.1 | >200 | >200 | >200 | >200 |
| CEF + βI | 1.6 | 1.6 | >200 | >200 | >200 | >200 |
| MCEF | >200 | >200 | >200 | >200 | 1.6 | 0.2 |
| MCEF + βI | **13** | 200 | >200 | >200 | **0.4** | **0.012** |

[a]MICs are given in µM, +; MHBCA, −; iron-deprived (ID)-MHBCA.
[b]βI β-lactamase inhibitor. _Klebsiella pneumoniae_ was tested with 4 µg/mL clavulanic acid, and _P. aeruginosa_ and _A. baumannii_ MICs were tested with 200 µg/mL of 3-aminophenylboronic acid (APB).
[c]Antibiotics: AMP ampicillin, BAMP bis-catechol-ampicillin, CEF cefaclor, MCEF mixed bis-catechol-mono-hydroxamate-cefaclor, LOR loracarbef, BLOR bis-catechol-loracarbef.
[d]Data in bold highlight a >fourfold reduction of the MIC when the antibiotic is combined with the β-lactamase inhibitor.

to the presence of enterobactin in the medium, which might compete against the mixed siderophore for iron chelation. However, the presence of enterobactin did not interfere with the action of BAMP and BLOR, both of which have a bis-catechol siderophore moiety. Finally, MICs were also measured for the _E. coli_ mutant lacking its β-lactamase (AmpC) (Table 2). The deletion of _ampC_ reduced the MICs at most by twofold for all antibiotics. Also, deletion of _ampC_ did not reverse the loss of activity of MCEF in ID-MHBCA as observed in the _entE_ mutant (Table 2). All strains evaluated in this study, except _E. coli_, express a β-lactamase that might impact the activity of conjugated and unconjugated β-lactams. To further understand how the enhanced activity of some SID-βL occurs, the susceptibility of conjugates to β-lactamases was investigated in more detail.

### Deletion or inhibition of β-lactamases has variable affects on the activity of conjugated β-lactams across species

To evaluate the impact of β-lactamases on conjugated and unconjugated β-lactams, MICs of _K. pneumoniae_, _P. aeruginosa_ and _A. baumannii_ were assessed in combination with a β-lactamase inhibitor (Table 3). All the strains used in this study naturally produce a β-lactamase: _K. pneumoniae_ ATCC 13883 expresses a class A penicillinase (SHV-1), _P. aeruginosa_ ATCC 27853 possesses an inducible chromosomal AmpC from the PDC-5 group and an OXA-396 from the OXA-50 group (atcc.org) and _A. baumannii_ ATCC 19606 possesses an inducible class C β-lactamase (ADC-158) and OXA-98 from the OXA-51-like group[30]. The β-lactamase inhibitor used was adjusted depending on the strain: clavulanic acid was used against the SHV-1 from _K. pneumoniae_, and 3-aminophenylboronic acid (APB) against the class C β-lactamases from _P. aeruginosa_ and _A. baumannii_. Results are presented in Table 3, where MICs shown in bold represent cases where the β-lactamase inhibitor reduced the MIC by at least fourfold.

The addition of clavulanic acid, against _K. pneumoniae_, decreased the MIC of AMP from >200 to 6.3 µM in both MHBCA and ID-MHBCA media (Table 3), but conjugate BAMP exhibited further enhanced activity with MICs decreasing from >200 to 1.6 µM in MHBCA and to 0.4 µM in ID-MHBCA. Clavulanic acid also reduced the MIC of MCEF against _K. pneumoniae_ from >200 µM to 13 µM, showing that MCEF is entirely dependent on β-lactamase inhibition for activity against this species. This increase of activity for MCEF in the presence of clavulanic acid was much less obvious in ID-MHBCA, with the MIC going from >200 µM to 200 µM (Table 3). As described earlier for _E. coli_ and based on evidence provided by the _entE_ mutant, the loss of activity of MCEF in iron-depleted conditions could be due to competition of MCEF against the endogenous siderophores produced by the bacterium.

Against _P. aeruginosa_, the presence of the β-lactamase inhibitor APB slightly reduced the MIC of BLOR from 13 to 6.3 µM in MHBCA, but much more markedly from >200 to 0.8 µM in ID-MHBCA (Table 3). Hence, for this species, the inactivity of BLOR in ID-MHBCA appears to be driven by β-lactamase activity. Conversely, BAMP did not need APB to be active, and its addition did not further improve its MIC against _P. aeruginosa_ or _A. baumannii_. In _A. baumannii_, APB further increased the activity of MCEF in ID-MHBCA (0.012 µM with APB compared to 0.2 µM without) (Table 3). However, _A. baumannii_, just like _P. aeruginosa_, remained resistant to the

**Fig. 2 | Kinetic parameters of purified β-lactamases. a** Schematic representation of the purification steps for each β-lactamase. SHV-1 was purified by cloning the respective gene of *K. pneumoniae* ATCC 13883 into an expression vector and were purified using a Ni-NTA column. AmpC (PDC-5) from *P. aeruginosa* ATCC 27853 was partly purified from an extract in its native form using an affinity column containing aminophenylboronic acid. **b** Individual data points of the Michaelis–Menten curves for SHV-1 (*n* = 3) and AmpC (*n* = 2) against BLOR and LOR. Each curve were done with at least seven points. **c** Mean and standard deviation of the kinetic parameters of SHV-1 and AmpC. The deviation for AmpC represents the gap between the two kinetic parameters measured. The asterisk, * indicates that the reported parameters were measured indirectly as a Ki against nitrocefin (NCF); ND not determined.

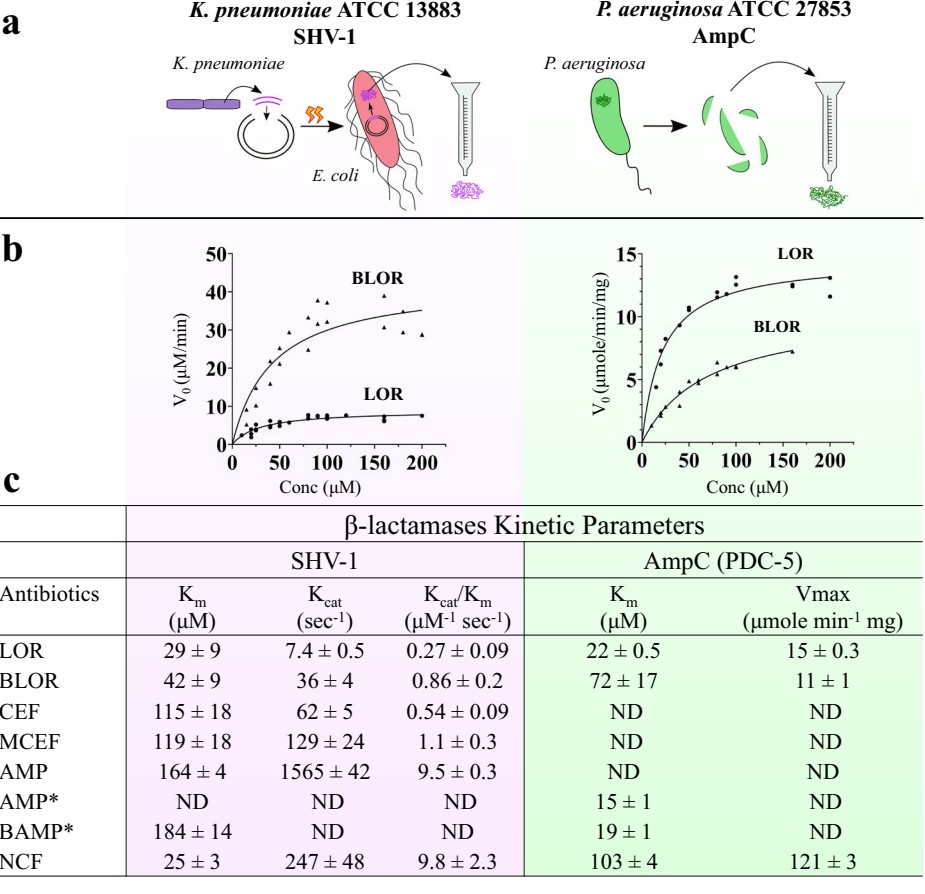

| β-lactamases Kinetic Parameters | | | | | |
|---|---|---|---|---|---|
| | SHV-1 | | | AmpC (PDC-5) | |
| Antibiotics | $K_m$ (μM) | $K_{cat}$ (sec$^{-1}$) | $K_{cat}/K_m$ (μM$^{-1}$ sec$^{-1}$) | $K_m$ (μM) | Vmax (μmole min$^{-1}$ mg) |
| LOR | 29 ± 9 | 7.4 ± 0.5 | 0.27 ± 0.09 | 22 ± 0.5 | 15 ± 0.3 |
| BLOR | 42 ± 9 | 36 ± 4 | 0.86 ± 0.2 | 72 ± 17 | 11 ± 1 |
| CEF | 115 ± 18 | 62 ± 5 | 0.54 ± 0.09 | ND | ND |
| MCEF | 119 ± 18 | 129 ± 24 | 1.1 ± 0.3 | ND | ND |
| AMP | 164 ± 4 | 1565 ± 42 | 9.5 ± 0.3 | ND | ND |
| AMP* | ND | ND | ND | 15 ± 1 | ND |
| BAMP* | 184 ± 14 | ND | ND | 19 ± 1 | ND |
| NCF | 25 ± 3 | 247 ± 48 | 9.8 ± 2.3 | 103 ± 4 | 121 ± 3 |

corresponding unconjugated β-lactam with MICs over 100 μM with or without APB.

### Siderophore conjugation slightly increased the catalytic efficiency of SHV-1 toward the conjugate

To further characterize the interaction of β-lactamases with SID-βL conjugates, SHV-1 from *K. pneumoniae* was purified, and AmpC from *P. aeruginosa* was extracted from a bacterial lysate (Fig. S2). OXA-98 from *A. baumannii* was also purified, and its kinetic parameters are compiled in Table S3. No hydrolysis by the class D β-lactamase OXA-98 from *A. baumannii* was detected for any of the cephalosporins (CEF, MCEF, LOR, and BLOR), and OXA-98 showed high $K_m$ values for AMP and BAMP (490 ± 12 and 499 ± 4 μM, respectively) (Table S3).

The $K_m$ values of SHV-1 (Fig. 2c) for the conjugated and unconjugated LOR or CEF were measured from three independent Michaelis–Menten curves. The difference in the $K_{cat}$ constant was more impactful on the final catalytic efficiency measured ($K_{cat}/K_m$). In fact, the higher $K_{cat}$ determined for BLOR led to approximately a threefold increase in the $K_{cat}/K_m$ for BLOR (0.86 ± 0.2 μM$^{-1}$ s$^{-1}$) than that measured for LOR (0.27 ± 0.09 μM$^{-1}$ s$^{-1}$). As for MCEF compared to CEF, the higher $K_{cat}$ determined for MCEF resulted in a twofold difference in the catalytic efficiency (1.1 ± 0.3 μM$^{-1}$ s$^{-1}$) compared to that found for CEF (0.54 ± 0.09 μM$^{-1}$ s$^{-1}$). Among LOR, BLOR, CEF, and MCEF, the highest $K_{cat}/K_m$ ratio was found for MCEF, which was also the only SID-βL showing an increased activity in combination with a β-lactamase inhibitor (Table 3).

The catalytic efficiency of SHV-1 against AMP was the highest (9.5 ± 0.3 μM$^{-1}$ s$^{-1}$), explaining the need to add clavulanic acid for AMP to be active against *K. pneumoniae* (Table 3). BAMP hydrolysis could not be detected with a spectrophotometer, and the $K_m$ was instead indirectly measured as a $K_i$. This method did not allow measurements of $K_{cat}$ and $K_{cat}/K_m$ for BAMP. Nevertheless, the $K_m$ measured for BAMP (184 ± 14 μM) was in the same order of magnitude as the $K_m$ measured for AMP using the

Michaelis–Menten curves (164 ± 4 μM) (Fig. 2c). Since the addition of clavulanic acid strongly potentiated both BAMP and AMP against *K. pneumoniae* (Table 3), the susceptibility of BAMP to hydrolysis from SHV-1 was strongly implied, even if the $K_{cat}$ was not measured.

### The $K_m$ measured with AmpC (PDC-5) hints toward a small decrease of the conjugated β-lactam affinity

The kinetic results for AmpC (PDC-5) extracted from *P. aeruginosa* are presented as the mean of two independent Michaelis–Menten curves, and the deviations represent the difference between the means of the two values (Fig. 2c). Foremost, the measured $V_{max}$ of AmpC was similar for LOR and BLOR. However, the $K_m$ for BLOR (72 ± 17 μM) was approximately threefold higher than that determined for LOR (22 ± 0.5 μM). This difference could partly explain the improved MIC (Table 1) measured for BLOR (13 μM in MHBCA) compared to LOR (MIC >200 μM), but it is unlikely to be the sole determinant. Indeed, the β-lactamase inhibitor APB only potentiated BLOR and not LOR against *P. aeruginosa* (Table 3), and this suggests that only the conjugate BLOR is able to cross the outer membrane and reach the PBPs. The $K_m$ ($K_i$) of AmpC for BAMP (19 ± 1 μM) and AMP (15 ± 1 μM) were also comparable, hinting at a similar affinity toward the β-lactamase (Fig. 2c). The presence of a β-lactamase inhibitor did not further improve the MIC of BAMP against *P. aeruginosa* (1.6 μM in MHBCA), and only slightly improved the MIC of AMP (from >200 to 100 μM) (see Table 3). For both BAMP and AMP, the β-lactamase AmpC does not seem to be impactful, neither in the efficiency of BAMP nor in the inactivity of AMP.

### Conjugated and unconjugated β-lactams are not significantly affected by efflux pumps in *E. coli* and *P. aeruginosa*

Efflux was investigated as a potential factor influencing the efficacy of conjugated β-lactams, either by enhancing their effectiveness or contributing to their lack of activity. To quantify the impact of efflux pumps

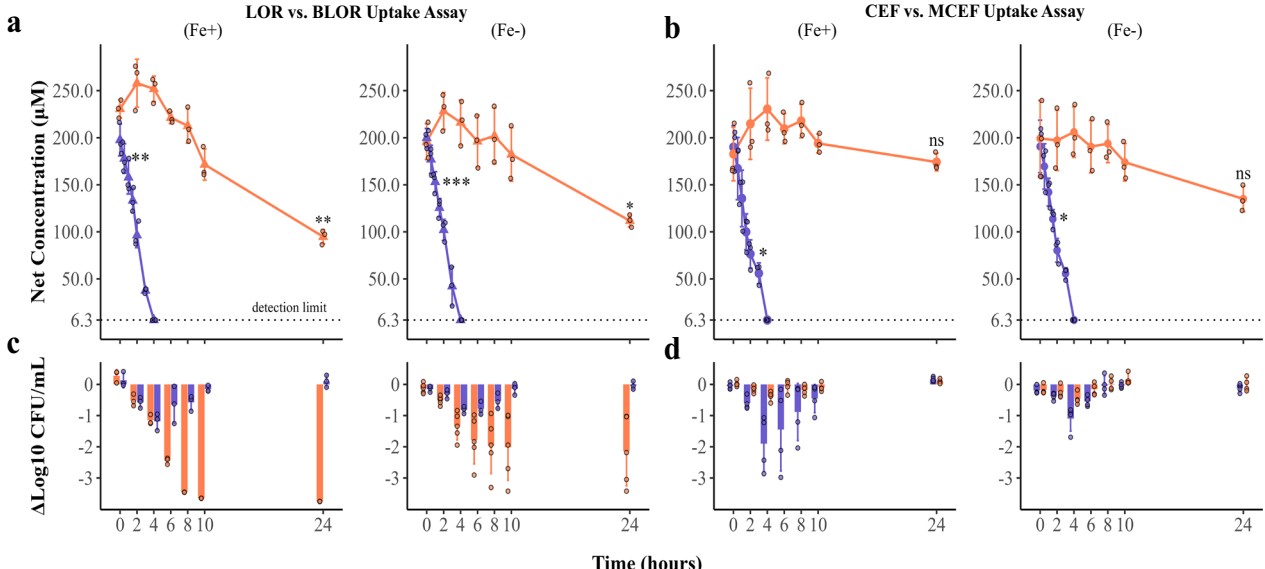

**Fig. 3 | Antibiotic uptake assay with *E. coli* MC4100. a, b** Curves represent the net concentration in culture supernatants of **a** loracarbef (LOR) (▲) or bis-catechol-loracarbef (BLOR) (▲) and of **b** cefaclor (CEF) (●) or mixed bis-catechol-mono-hydroxamate-cefaclor (MCEF) (●), in MHBCA (Fe+) and iron-deprived (ID)-MHBCA (Fe-), over a 24-h period. Net concentrations represent the levels of each antibiotic measured in the presence of *E. coli*, corrected by subtracting the corresponding concentrations measured in the absence of bacteria at each time point. Normality was verified with the Shapiro–Wilk's test and the statistical analysis is a comparison of repeated measures ANOVA between each time point relatively to the 0-h time point ($n$ = 3). BLOR showed a statistically significant uptake at 24 h (*$p \le 0.05$, **$p \le 0.01$, ***$p \le 0.001$), but MCEF uptake was not significant (ns). The Δlog10 CFU/mL (bars with individual data points) was calculated by subtracting the bacterial count with drugs to the bacterial count without drugs (normal growth) at each time point. Panel **c** is the Δlog10 CFU/mL of LOR (blue) and BLOR (orange), and **d** is the Δlog10 CFU/mL of CEF (blue) and MCEF (orange).

on the activity of the antibiotics, MICs were tested in combination with the non-specific efflux pump inhibitor phenylalanine-arginine-β-naphthylamide (PAβN). PAβN has been optimized to counteract *P. aeruginosa* efflux pumps. In this species, the addition of 25 μg/mL of PAβN potentiated the activity of BAMP by 8-fold. However, it antagonized the activity of all conjugates in other species, leading to an inhibition of activity against *A. baumannii* (see results in Table S4). On the other hand, the control antibiotics used in this assay (rifampicin and erythromycin) saw their activity improved in the presence of PAβN, possibly due to a combination of efflux pumps inhibition and PAβN membrane permeabilization, as previously suggested[31]. As such, a possible explanation for the loss of activity of conjugates in the presence of PaβN (opposite effect to that observed for rifampicin and erythromycin) might be a secondary target affecting the active transport of side-rophores through TBDTs. However, further tests are needed to confirm this hypothesis, and no conclusions could be drawn. To complement the PAβN assays, a variety of efflux pump mutants of *E. coli* and *P. aeruginosa* (see Table S5 for details) were analyzed.

As expected, the deletion of *acrAB* in *E. coli* reduced the MIC of the control antibiotic erythromycin from 128 to 4 μg/mL (32-fold). However, the MICs of AMP, CEF and LOR and their siderophore conjugate counterparts were not affected by more than twofold (Table S5). Similarly, in *P. aeruginosa*, neither the double deletion of Δ*mexCD-oprJ*/Δ*mexEF-oprN*, or Δ*mexAB-oprM*/Δ*mexEF-oprN*, nor the overexpression of *mexAB*, *mexCD*, and *mexEF*, affected the activity of BAMP, BLOR and MCEF by more than twofold. In contrast, the MIC of the control drug levofloxacin was significantly altered in some of these mutants. Against the double mutant Δ*mexAB-oprM*/Δ*mexEF-oprN*, the MIC of levofloxacin dropped to ≤0.06 μg/mL, compared to 0.5 μg/mL for the parent strain. Also, when all pumps were overexpressed (strain *P. aeruginosa* 2302), the MIC of levofloxacin rose to 16 μg/ml. Hence, the activity of efflux pumps did not explain the difference in the antibiotic activity observed for some conjugated compared to the unconjugated β-lactams against these bacterial species. To further understand the enhanced activity of the conjugates, drug uptake was next investigated in more detail.

## The hyperpermeable *E. coli* strain, *lptD*4213, potentiates MCEF without impacting the activity of BAMP or BLOR

The *E. coli* mutant *lptD*4213 (imp4213) is defective in lipopolysaccharide export[32], which allows large antibiotics like vancomycin to cross the outer membrane barrier and reach its periplasmic target (MIC of 0.5 instead of >128 μg/ml for the parental strain MC4100). MICs from Table S6 shows that LOR and BLOR did not benefit from the hyperpermeability of the *lptD* mutant strain (no more than a twofold change in MIC between the mutant and parent strains). On the other hand, the MIC of MCEF was significantly reduced against the *lptD*4213 mutant (1.6 μM), falling even below that of CEF (3.1 μM). Unlike the observations with MC4100 in MHBCA, no growth of *lptD*4213 was detected at high MCEF concentrations (i.e., no Eagle effect), and its MIC remained at 1.6 μM. Overall, these results suggest that MCEF transport across the outer membrane of *E. coli* wild-type is limited in MHBCA and, in ID-MHBCA, its activity is counteracted in part by enterobactin (see previous results with Δ*entE* from Table 2).

Inversely, AMP (but not BAMP), showed a lower MIC for the hyperpermeable strain *lptD*4213 (0.4 vs. 13 μM) (Table S6). This highlights the fact that the outer membrane of *E. coli* limits the action of AMP. Since BAMP is equally active against both strains (parent and *lptD*), the conjugate does not appear to be affected by the same type of outer membrane restrictions in *E. coli*, as also suggested by its dependence on TBDTs when compared to AMP (Table 2).

## The antibiotic uptake assay allows a comparison between the transport of conjugated and unconjugated β-lactams in *E. coli*

The net uptake of LOR versus BLOR and CEF versus MCEF in *E. coli* MC4100 were assessed indirectly by measuring the residual drug concentrations in culture supernatants using ultra-performance liquid chromatography-mass spectrometry (UPLC-MS) over a 24-h period, at various time points. By controlling drug aqueous stability in the absence of bacteria, this assay assumes that the decreasing drug concentration in the supernatant results, in major part, from cellular uptake and subsequent binding to PBPs, preventing the release of intact drug back into the supernatant. For experiments reported in Fig. 3a, b, all drugs were tested at

an initial concentration of 200 µM combined with a bacterial inoculum of 8.5 log10 CFU/mL, allowing a wide range of measurable concentrations by UPLC-MS over time (Fig. 3a). Additionally, at each time point, bacterial CFU/mL were determined and compared to normal growth without drugs (Fig. 3c, d). This bacterial count enabled correlation of drug uptake with the bactericidal effect.

## Adding a siderophore portion reduced the net uptake of β-lactams

The net concentration of BLOR in the medium significantly decreased in MHBCA and ID-MHBCA after 24 h in the presence of *E. coli* MC4100 (Fig. 3a). In contrast, LOR concentration fell below the UPLC-MS detection limit during the first 4 h of the assay either in MHBCA and ID-MHBCA. LOR concentrations were significantly different from the 0-h time point after 0.5 h in MHBCA and after 1.5 h in ID-MHBCA. Moreover, the concentration of CEF also fell below the detection limit during the first 4 h in MHBCA and ID-MHBCA, and a significant decrease was observed after 2 and 1.5 h, respectively (Fig. 3b). However, MCEF concentration levels were never significantly different from the 0-h time point in MHBCA or ID-MHBCA.

## Less BLOR uptake is needed to reach the same bactericidal effect of LOR

Figure 3c, d shows the difference in bacterial counts between the growth control without antibiotic and the counts in the presence of the conjugated or unconjugated β-lactams (Δlog10 CFU/mL). At the 4-h time point in MHBCA, the level of bacterial inhibition is approximately the same for LOR and BLOR (~1.2 log10 CFU/mL less than the control without antibiotic). However, at that time point, the net uptake of BLOR remained low and not significantly different from the 0-h data, while over 194 µM of LOR had already disappeared from the supernatant. This suggests that, for equivalent antibacterial activity against *E. coli*, significantly less conjugated drug is required in this assay. After the 4-h time point, BLOR continues to inhibit bacterial growth efficiently, whereas LOR disappearance from the medium allows cells to regrow (Fig. 3c). Surprisingly, in ID-MHBCA, the bactericidal effect is reduced for both LOR and BLOR (Fig. 3c [Fe-]). It did not, however, alter significantly its antibacterial activity, as its MIC was twofold lower in ID-MHBCA (0.006 µM) compared to MHBCA (0.012 µM) (Table S6).

## MCEF limited uptake in MHBCA and in ID-MHBCA correlates with its lack of bactericidal activity

Unlike LOR compared to BLOR, CEF was more effective than MCEF at killing *E. coli* (Fig. 3d), consistent with its lower MIC against *E. coli* MC4100 (Table S6). MCEF net concentrations in MHBCA or ID-MHBCA remained stable over time, indicating minimal or no uptake by *E. coli* MC4100, which explains its lack of inhibiting activity (Fig. 3d). Indeed, the improved MIC for MCEF against the hyperpermeable *lptD*4213 strain suggested that poor outer membrane penetration was a key limitation for antimicrobial activity (Table S6).

In contrast, CEF was taken up by bacteria and showed bactericidal activity (Fig. 3d). However, like LOR, CEF quickly disappeared from cultures in both MHBCA and ID-MHBCA, allowing bacterial regrowth with no net killing at 24 h under the high inoculum conditions used. Notably, CEF bactericidal activity was slightly reduced in ID-MHBCA, as observed for LOR and BLOR. Additional experiments will be required to explain the reduced bactericidal efficiency of antibiotics in ID-MHBCA, but a reduced metabolic activity due to iron depletion may be considered as an explanation.

## Metabolic byproducts had a limited impact on the concentrations measured during the uptake assay

To ensure that the drastic drops of the unconjugated β-lactams after 4 h were not due to degradation from secreted metabolites or other metabolic wastes, an additional drug stability control was done using the supernatant of the growth control without an antibiotic. Four-hour supernatants, with

and without iron, were collected by filtration, and the stability of LOR and CEF in those supernatants was measured after 3 h, which corresponds to the last time point before LOR and CEF concentrations fell below the detection limit. Compared to the drug stability measured in the medium without antibiotic and cells, the residual concentration of LOR or CEF did not decrease by more than 12% in the "used" supernatants (Table S7). This additional slight degradation does not, however, affect the conclusions derived from the assay as outlined below.

To summarize, the net uptake of the unconjugated drugs was found to be higher than that of conjugated β-lactams. However, despite this finding, even a small but steady uptake of conjugates appears sufficient to ensure efficient antibiotic action, as evidenced by low MIC values and increased bacterial elimination over time. This led us to investigate the relative binding of conjugated and unconjugated β-lactams for PBP targets.

## Conjugates have an increased affinity to PBP3 relative to unconjugated β-lactams

The relative affinity of conjugates for PBPs was measured by determining the apparent $IC_{50}$ in a competition assay against the fluorescent penicillin Bocillin FL (schematized in Fig. 4a). Uncropped gels can be seen in Fig. S3. Affinity referred to the $IC_{50}$ and should not be confused with the strength of the noncovalent interaction between the β-lactam and the PBP. The apparent $IC_{50}$ was defined as the drug concentration needed to block 50% of Bocillin FL binding to a specific PBP. All $IC_{50}$ values are shown in Table S8 and ratios determined between the values of the β-lactams, and their conjugated forms are reported in Fig. 4b. Interestingly, all conjugates showed improvement in their apparent $IC_{50}$ for PBP3. The ratios for PBP3 increased in all bacterial species, with a marked fold change of 199 recorded in favor of BLOR compared to LOR in *E. coli* (Fig. 4b, c). This important increase of affinity for BLOR toward PBP3 also applied to *K. pneumoniae*, *P. aeruginosa*, and *A. baumannii* PBP3 with a 46-, >16-, and 91-fold change, respectively (Fig. 4b). Similar increases in PBP3 relative affinities were observed with MCEF and to a lesser extent with BAMP compared to their respective unconjugated drugs.

The general increased binding of conjugated β-lactams for HMW PBPs (PBP1 to 3) was not seen for LMW PBPs (PBP4 to 6). No fold change was measured for PBP5/6 and fold changes towards the PBP4 of *E. coli* and *K. pneumoniae* did not exceed 4.7 or were even lower than 1, meaning that the siderophore causes a loss of affinity for this LMW PBP. PBP4 of *P. aeruginosa* was expressed at low levels (Fig. 4d). Although detectable on gels, its analysis was compromised due to fluorescence from the PBP5/6 band.

## The conjugation of a β-lactam with a siderophore can increase the range of targeted PBPs

In *P. aeruginosa*, LOR only had a measurable $IC_{50}$ against PBP1b (1.8 ± 0.5 µM, Table S8). In contrast, BLOR generated measurable $IC_{50}$ for multiple targets: 0.58 ± 0.04 µM for PBP1a, 1.4 ± 0.4 µM for PBP1b, 4.8 ± 3.0 µM for PBP2, and 3.2 ± 0.1 µM for PBP3. Conjugating LOR or CEF to a siderophore also enabled a measurable $IC_{50}$ for PBP2 in *K. pneumoniae* and *P. aeruginosa* (Table S8).

Notably, MCEF gained a higher $IC_{50}$ for *E. coli* PBP1a/1b, becoming the preferred target over PBP3 ($IC_{50}$ of 0.15 ± 0.01 µM vs. 0.32 ± 0.01 µM, respectively). In contrast, CEF preferred PBP3 over PBP1a/1b ($IC_{50}$ of 15.4 ± 0.9 µM vs. 34 ± 2 µM, respectively). However, the twofold difference in $IC_{50}$ between PBP1a/1b and PBP3 for both CEF and MCEF may not be biologically significant. Overall, MCEF exhibited a 227-fold increase in $IC_{50}$ for PBP1a/1b and a 49-fold increase for PBP3 compared to CEF (Fig. 4b).

## BAMP and AMP had the same relative affinities for the PBPs of *P. aeruginosa*

For AMP and BAMP, conjugation of AMP to the siderophore did not enhance any $IC_{50}$ value by more than 1.5-fold. Also, a ratio of 0.28 was obtained for PBP1b, indicating a lower affinity of BAMP (Fig. 4b). Among all the tested conjugated and unconjugated β-lactams, AMP and BAMP exhibited the highest binding affinity towards PBP3. However, siderophore

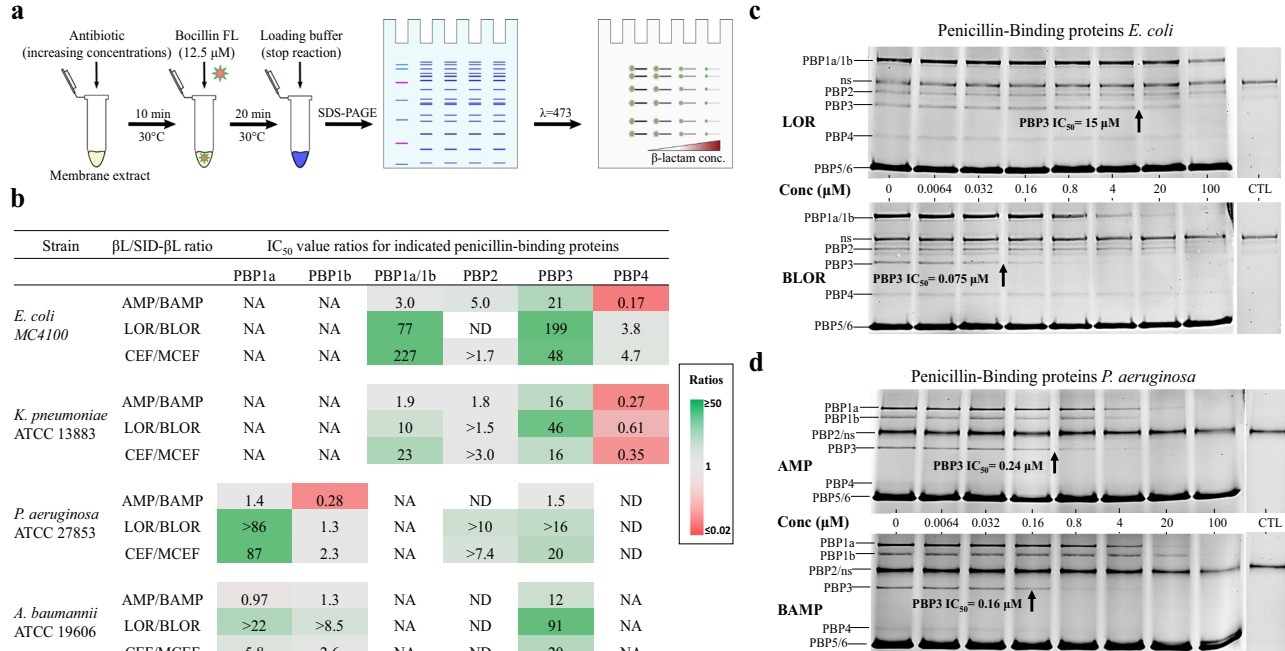

**Fig. 4 | PBP binding of conjugated (SID-βL) compared to unconjugated (βL).**
**a** Schematic protocol for measurement of apparent IC$_{50}$ values using bacterial membranes as the source of PBPs. **b** Apparent IC$_{50}$ value ratios (calculated from IC$_{50}$ values provided in Table S8) for PBPs 1 to 4 of the different bacterial species tested. The ratios (βL/SID-βL) are represented in a heat map with the intensity ranging from red (SID-βL loses affinity compared to βL) to green (SID-βL gained affinity compared to βL). NA not applicable, ND not determined. Gel images **c**, **d** show the

fluorescent PBP profiles on SDS-PAGE obtained from *E. coli* MC4100 and *P. aeruginosa* ATCC 27853 membrane preparations, respectively. The arrow points to the apparent PBP3 IC$_{50}$ value measured with the Quantity One software (4.6.6). When two PBPs overlapped on gels, the IC$_{50}$ were measured together as one entity (e.g., *E. coli* or *K. pneumoniae* PBPs 1a and 1b). The concentrations of test drugs in each lane are given in μM and the CTL lane represents a membrane aliquot without test drug or Bocillin FL to highlight fluorescent proteins that are not PBPs (ns).

conjugation did not further enhance their affinity, unlike the improvements observed with BLOR and MCEF. The observed significant difference between the AMP/BAMP pair (IC$_{50}$ ratio of 1.5) and the LOR/BLOR pair (ratio >16) was investigated in the next section using the purified PBP3 of *P. aeruginosa* (PaPBP3) as the target for these drugs.

### The increased affinity for PaPBP3 of BLOR compared to LOR is confirmed when using the purified target

A soluble version of the PBP3 of *P. aeruginosa* was purified using an His-tag at the C-terminal. The IC$_{50}$ determination assay was done using 150 ng of PaPBP3 and 500 nM Bocillin FL against increasing concentrations of the tested drugs. In Fig. 5, the IC$_{50}$ values and the IC$_{50}$ ratios (βL /SID-βL) obtained with the purified PaPBP3 are compared to the IC$_{50}$ values obtained with the bacterial membrane fraction (see uncropped gels in Fig. S3). The IC$_{50}$ AMP/BAMP ratio was 0.97 with purified PaPBP3 and 1.5 with the membrane extract, confirming a similar affinity of AMP and BAMP for *P. aeruginosa* PBP3. In contrast, the IC$_{50}$ LOR/BLOR ratio was 23 using the purified PaPBP3 or >16 with the membrane fraction, highlighting in both tests a significant increase in BLOR affinity for PaPBP3 compared to LOR.

To summarize, investigations on the interaction between SID-βL and PBPs revealed that the addition of a siderophore to a β-lactam can generally have a beneficial effect on its affinity for the HMW PBPs and even broaden the number of primary PBP targets. This unforeseen property of SID-βL conjugates can certainly contribute to their enhanced activity compared to unconjugated β-lactams.

### Discussion
The main objective of this study was to understand how siderophores can enhance the activity of β-lactams against Gram-negative bacteria. Synthetic SID-βL conjugates, as well as the natural sideromycins, act as Trojan horses to facilitate their uptake through the Gram-negative outer membrane using bacterial TBDTs as doors. Therefore, their antibacterial activity should be

proportional to the expression level of siderophore transport systems, which would be maximized in iron-restricted environments. The MIC data of SID-βL conjugates and unconjugated β-lactams across various bacterial species grown in iron-rich and iron-depleted media revealed additional complexities and perspectives related to the mode of action of SID-βL. To understand the various factors modulating the antibacterial activity of SID-βL, four mechanistic aspects were studied: susceptibility to β-lactamases, efflux pumps, uptake across the outer membrane and relative affinity to PBP targets. Table S9 summarizes the findings of this study by reporting the cellular factors and their relative contribution depending on the bacterial species.

Remarkably, the increased binding of conjugated β-lactam towards the HMW PBPs was a major factor across all species that correlated with their enhanced efficiency. In *E. coli*, this sole factor mostly explained the enhanced activity of BLOR compared to LOR. While an increased binding for some conjugated β-lactams for PBPs has also been noted in the literature[33,34], the present work highlights the important role of this characteristic in the mode of action of β-lactam-based sideromycins against several bacterial species. As a comparison, an 11-fold decrease in the IC$_{50}$ for *E. coli* PBP3 was reported for cefiderocol compared to ceftazidime[33]. Similarly, assays using purified PBP3 from *P. aeruginosa* and mass spectrometry analysis confirmed that cefiderocol increased affinity for this PBP compared to ceftazidime, although the enhancement was less pronounced than that observed for *E. coli*[35]. This increased affinity with siderophore conjugates was not observed in a study on the siderophore-monocarbam, MC-1, in *P. aeruginosa*[36]. This may indicate that siderophore-cephalosporin conjugates, like BLOR and MCEF, are more prone to gain PBP affinity compared to siderophore-monocarbam or siderophore-penicillin like BAMP, as seen in this study.

This discrepancy in the binding of cephalosporins and penicillins having a similar acylamino chain has been explored by ref. 37. Against the PBP2 of the resistant *Neisseria gonorrhoaea* H041, cefoperazone

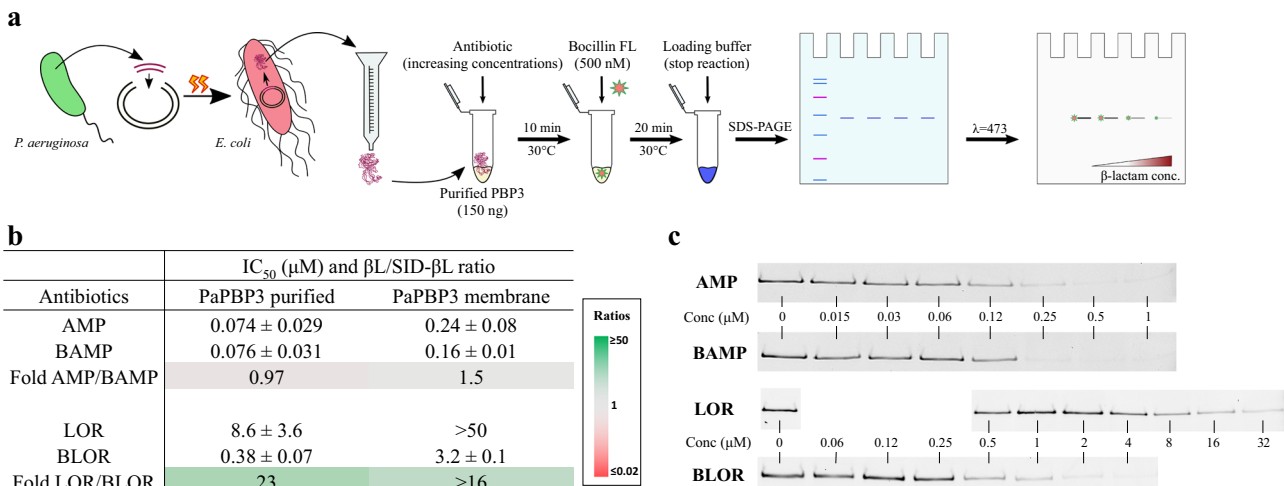

**Fig. 5 | β-lactams and conjugated β-lactams IC$_{50}$ and ratios values for purified *P. aeruginosa* PBP3 (PaPPB3). a** Schematic representation of the cloning and expression of PaPBP3 gene in *E. coli*. Purification was done using a Ni-NTA column. The purified PBP3 from *Pseudomonas aeruginosa* ATCC 27853 (PaPBP3) was used in the PBP assay with the test antibiotics and Bocillin FL. The IC$_{50}$ is the concentration of the test antibiotic needed to prevent the binding of 50% of Bocillin FL to PaPBP3. **b** Means and standard deviation of IC$_{50}$ values (uM) (*n* = 3) and IC$_{50}$ ratios for βL/SID-βL for *P. aeruginosa* PBP3 (either purified or in the membrane preparation). The ratios (βL/SID-βL) are represented in a heat map with the intensity ranging from red (SID-βL loses affinity compared to βL) to green (SID-βL gained affinity compared to βL). **c** Examples of assays with PaPBP3 on SDS-PAGE comparing a set of AMP concentrations to BAMP and of LOR concentrations to BLOR.

(a cephalosporin) had an increased affinity for the precovalent state compared to the structurally analogous, more potent, penicillin (i.e., piperacillin). The latter has a faster acylation rate. In such cases, penicillins rapidly equilibrate between their solution state and the binding pocket, leading to quicker covalent bond formation (acylation). In contrast, cephalosporins exhibit a slower equilibrium, which extends the duration of the pre-acylation state. For conjugates, the increased molecular size likely prolongs this pre-acylation equilibrium further, as additional pharmacophore alignment is required prior to covalent bond formation. As a result, the inhibition of PBPs by such conjugates may be driven more by electrostatic interactions than by actual acylation, if covalent bonding occurs at all. Thus, conjugating cephalosporins, which already display a strong electrostatic component in the precovalent state, further enhances this characteristic, increasing overall inhibition. On the other hand, conjugating a siderophore to a penicillin, which might rely more on rapid acylation, may slow acylation but compensate with enhanced electrostatic interactions, ultimately maintaining a similar inhibitory potency.

Regarding drug uptake, it is generally accepted that β-lactams passively diffuse through the outer membrane or the water-filled porins following the concentration gradient to reach the periplasmic space of *E. coli*[17]. Instead, SID-βL uses TBDTs for an active and specific uptake, as seen with the loss of activity in the *E. coli* Δ*tonB* or specific TBDT mutants (Table 2). This prompted us to measure the net concentrations of drugs in bacterial culture supernatants over time to compare the uptake of LOR and BLOR (Fig. 3a, b). The uptake assay in *E. coli* showed a reduced net entry of β-lactams conjugated to a siderophore compared to the β-lactams alone over a 24-h period. However, this does not necessarily correlate to a reduced activity. On the contrary, BLOR showed prolonged bacterial inhibition over time compared to LOR (Fig. 3c, d). A limitation of this assay is the fact that antibiotic molecules retained in the membrane might not be measured but bacterial counts mitigated this effect by correlating the killing with the residual supernatant concentration over time.

Finally, if the outer membrane was a limiting factor in *E. coli*, the use of the hyperpermeable *E. coli* strain *lptD*4213 should have potentiated the activity of the antibiotics tested. An increased activity was indeed seen with MCEF and AMP confirming their limited uptake. However, the MICs were unchanged for LOR, CEF, BLOR and BAMP. This points out the strong possibility that the enhanced activity of BLOR against *E. coli* compared to LOR is ultimately and effectively due to the increased binding of the SID-βL

conjugate to multiple HMW PBPs (Fig. 4). Accordingly, the greater activity of BAMP compared to AMP can be explained by two factors: improved uptake and affinity. For MCEF, even if an increased binding to PBP3 was measured, the limited uptake in MHBCA and the Eagle effect, mostly caused by enterobactin expression (Table 2) in ID-MHBCA, prevented its activity. A study in *P. vulgaris* measuring the impact of the Eagle effect seen in vitro showed that high β-lactamase induction also happened in vivo, leading to reduced survival of infected mice at higher doses of cefmenoxime[38]. This could be of importance for siderophore-β-lactam conjugates inducing this phenomenon as a small dose might be more efficient than high doses to treat infections.

While the impact of relative uptake of SID-βL conjugates is mitigated in *E. coli*, this factor is crucial in *P. aeruginosa*. It is well established that *P. aeruginosa* and *A. baumannii* limit the diffusion of many molecules due to their selective porins, namely OprF for *P. aeruginosa* and OmpA for *A. baumannii*[39,40]. In *P. aeruginosa*, the uptake of antibiotics is mostly porin-independent and specific molecular characteristics, like positive charge, are required for self-promoted uptake[41,42]. Remarkably, a mutant strain lacking 40 identified porin genes shows no MIC difference across a broad set of antibiotics compared to the wild-type strain[43]. Moreover, it has been shown that CEF has a very slow uptake in *P. aeruginosa*[44]. It is thus possible to hypothesize that the intrinsic resistance to AMP, CEF and LOR of *P. aeruginosa* and *A. baumannii* might be due to a limited transport relative to any other factors. In those cases, conjugation to a siderophore might considerably enhance entry and activities of β-lactams, as seen with BAMP and BLOR. Performing the uptake assay with a β-lactamase Δ*ampC* mutant of *P. aeruginosa* could allow us to answer this question. Furthermore, in these species, a low iron concentration in the medium would probably increase the uptake of the conjugates. Indeed, it was previously demonstrated that the uptake of [thiazole-$^{14}$C]-cefiderocol was increased in ID-MHBCA using a 10-min assay[21]. Additionally, the lower MICs of BAMP and BLOR that were observed in combination with a β-lactamase inhibitor in ID-MHBCA, compared to MHBCA (Table 1), support this hypothesis.

The results obtained to evaluate the importance of resistance factors like β-lactamases and efflux pumps were helpful to explain intrinsic resistance, but did not account for the enhanced efficacy of conjugated β-lactams compared to their unconjugated forms. For example, the resistance of *K. pneumoniae* to AMP and MCEF could be attributed to the catalytic efficiency ($K_{cat}/K_m$) of its β-lactamases, as well as the observed activity increase

when these antibiotics were used in combination with clavulanic acid. Additionally, further studies investigating different efflux pumps, might be needed. As an example, MuxABC-opmB in *P. aeruginosa*, has been identified to interfere with the action of cefiderocol by increasing the excretion of the endogenous siderophore pyoverdine[45].

β-Lactamases are important factors to consider in the future development of β-lactams, particularly SID-βL conjugates. In the case of BAMP, BLOR, and MCEF, the siderophore conjugation did not provide protection against these enzymes. However, a reduced susceptibility to β-lactamases has previously been reported when a siderophore is attached at the C3 position, as observed in the case of cefiderocol[46]. Carbapenems or monobactams could also be interesting β-lactams to conjugate due to their reduced susceptibility to β-lactamases. A tri-catechol siderophore linked to meropenem has been described and shows great activity against *E. coli*[47]. A bis-catechol-aztreonam is currently under investigation[48]. Besides, by conjugating a siderophore at the C7-α-position of cephalosporin could be an interesting avenue to improve the stability of future conjugates against β-lactamases (e.g., as in cefoxitin)[49]. This attribute, combined with the tendency of cephalosporins to exhibit increased affinity for HMW PBPs, could offer a promising combination for future conjugates.

Overall, this study systematically examined cellular factors that may influence the activity of β-lactams conjugated to siderophores in different Gram-negative bacterial species. The direct comparison between conjugates and their unconjugated β-lactam counterparts highlighted key factors explaining the enhanced activity of SID-βL conjugates depending on the species. Their steady uptake associated with a prolonged inhibition and their increased affinity towards HMW PBPs were the major findings of this work. Even if other factors might play a role in their activity, future rational design of β-lactam-based sideromycins should capitalize on these beneficial properties to create narrow or broad-spectrum antimicrobials that could overcome existing resistance barriers.

## Methods

### Antibiotic preparation and compounds

Syntheses and characterization of all three SID-βL conjugates (bis-catechol-ampicillin [BAMP], bis-catechol-loracarbef [BLOR], and mixed bis-catechol-mono-hydroxamate-cefaclor, [MCEF]), as well as loracarbef (LOR) are described in the supplemental information. Ampicillin (AMP), cefaclor (CEF), clavulanic acid, 3-aminophenylboronic acid, and nitrocefin (NCF) were purchased from Sigma-Aldrich (Oakville, ON, Canada). Conjugated and unconjugated β-lactams were weighed and dissolved first in a half volume of potassium phosphate (KP) buffer 0.05 M pH 8 and then half volume KP buffer 0.05 M pH 6 to obtain a final concentration of 5 mg/ml.

### Growth conditions and iron-deprived medium

All bacterial strains (see description and sources Table S1) were stored at −80 °C, and grown on tryptone soya agar (TSA) at 35 °C. The iron-deprived (ID)-MHBCA (cation-adjusted Muller-Hinton Broth, Becton Dickinson, Mississauga, ON, Canada) medium was prepared as reported by refs. 21,50. Briefly, MHBCA was treated with 10% Chelex® 50-100 mesh (Sigma) and stirred for 6 h at room temperature. The resin was removed by filtration. Divalent $Ca^{2+}$ ($CaCl_2$), $Mg^{2+}$ ($MgCl_2$), and $Zn^{2+}$($ZnSO_4$), were then added in the iron-depleted medium at 22, 12, and 1 µg/ml respectively. Finally, the pH was adjusted to 7.3, and the medium was sterilized with a 0.2-µm filter. Regular MHBCA (i.e., not treated with Chelex®) was used to compare the inhibitory activity of conjugates in iron-sufficient conditions. The resin used for Chelex treatment was always fresh and had never been regenerated, ensuring consistent iron removal.

### Growth kinetics in ID-MHBCA and MHBCA

To confirm that iron was the growth limiting factor in ID-MHBCA, growth curves were produced in the presence of different concentrations of supplemental iron. ID-MHBCA was supplemented with $Fe^{3+}$ (from $FeCl_3$) at 0.01, 0.1 and 1 µg/mL ($1.8 \times 10^{-7}$, $1.8 \times 10^{-6}$, and $1.8 \times 10^{-5}$ M). Each condition was tested in technical duplicates and biological duplicates in 96-well

plates. The pH was confirmed to be between 7 and 7.5 for every condition. Bacterial inoculum was adjusted to $5 \times 10^5$ CFU/mL and added to the wells containing the different iron concentrations. ID-MHBCA and MHBCA with or without inoculation were used as controls. Growth was monitored with $OD_{600}$ at 0, 6, 12, and 24 h with an Epoch microplate reader (Biotek Instrument Inc., Vermont). All results were compiled in GraphPad Prism version 10.2.2. The average $OD_{600}$ was compared to the growth observed in the MHBCA control condition. Normality was verified with the Agostino-Pearson test for the dataset at each time point. The statistical difference was measured by a two-way ANOVA, and a Dunnett multiple comparison tests were conducted to confirm statistical significance between MHBCA and other conditions.

### Antibiotic susceptibility testing

Minimal inhibitory concentrations (MICs) were performed using a broth microdilution technique according to the Clinical and Laboratory Standards Institute (CLSI) guidelines[51]. MICs were determined in MHBCA and in ID-MHBCA. To check the impact of β-lactamases, MICs in the presence of a β-lactamase inhibitor were also determined. The inhibitor was homogenized at 2X the final concentration and added to the bacterial inoculum just before distributing cells into the wells containing the drug dilutions. The final concentration of each inhibitor was 4 µg/mL for clavulanic acid (tested against *K. pneumoniae*) and 200 µg/mL for 3-aminophenylboronic acid (tested against *P. aeruginosa* and *A. baumannii*).

### *P. aeruginosa* lysis and AmpC expression

The native form of the β-lactamase AmpC was purified from *P. aeruginosa* ATCC 27853. Briefly, 1 L of culture was incubated with gentle shaking (100 rpm) at 37 °C, and then exposed to 0.25×MIC of imipenem (Sigma) once the $OD_{600}$ reached 0.2 to induce AmpC expression. Growth was stopped in the late exponential phase ($OD_{600}$ ~0.9) by a centrifugation step at 3500 rpm for 15 min to collect the cell pellet. The pellet was washed with a potassium phosphate (KP) buffer 0.05 M pH 7 (20 mL final suspension volume) and treated with lysozyme (400 µg/mL) for 1 h at 37 °C with gentle agitation. DNase and RNase (Qiagen) at 10 µg/mL and a protease inhibitors cocktail (P8465, Sigma) at 200 mg/20 mL of cell suspension were added and incubated for another 30 min at room temperature. Cells were disrupted with a French pressure cell adjusted to 20 000 psi. Debris were removed with a $10,000 \times g$ centrifugation for 10 min, and the supernatant was ultra-centrifuged at $100,000 \times g$ for 45 min at 4 °C. The supernatant containing the AmpC enzyme was then flash frozen and kept at −80 °C.

### *P. aeruginosa* AmpC purification

The purification of AmpC from the culture supernatant obtained above was conducted as described in the Catwright and Waley protocol[52] using an Affi-gel 10 matrix (Bio-Rad, Hercules, CA, USA) loaded in a column (G-Bioscience, St-Louis, MO, USA). First, the resin was washed with cold deionized water. The resin was then collected and mixed with 20 mL of the 3-aminophenylboronic acid solution (1 M $KHCO_3$ containing 2 g of 3-aminophenyl boronic acid and 2 g of sorbitol) at 4 °C for 1 h, and the pH was maintained at 8 with solid $KHCO_3$. The mixture was loaded onto the column and washed with 10 column volumes of 1 M NaCl/0.5 M sorbitol (pH 7), followed by 10 column volumes of 0.5 M borate (pH 7), and then 10 column volumes of the loading buffer (20 mM triethanolamine HCl/0.5 M NaCl, pH 7). The sample was dialyzed in the loading buffer before being loaded onto the column. The column was washed with the loading buffer until no absorbance was measured at $OD_{280}$. AmpC was eluted in fractions of 1 mL with 0.5 M borate/0.5 M NaCl pH 7. Each fraction was tested with 100 µM of nitrocefin to visualize the elution peak. Fractions demonstrating activity were concentrated with an Amicon® ultra-centrifugal filter unit (MilliporeSigma) with a 10 kDa cut off. The purity was visualized on a 10% bis-acrylamide gel. Only pure fractions showing a single band at 40 kDa were pooled together and then dialyzed at 4 °C in a KP buffer 0.05 M pH 7 (see Fig. S2). The protein concentration was determined with a Bradford protein reagent (Bio-Rad).

## PBP3, SHV-1, and OXA-98 cloning and overexpression

The PBP3, encoded by the gene *ftsI*, (residues 53 to 579) from *P. aeruginosa* ATCC 27853 (PaPBP3), and the β-lactamases SHV-1 (residues 23–286) from *K. pneumoniae* ATCC 13883 and OXA-98 (residues 33-274) from *A. baumannii* ATCC 19606, were amplified using primers with flanking regions for a Gibson assembly[53] (see primers Table S10). GeneArt Gibson Assembly Hifi master mix (Thermo Fisher Scientific) was used for the cloning. All proteins were cloned into the linearized vector pET-24b(+) system leading to a C-terminal His$_6$ and transformed into electrocompetent cells of strain *E. coli* BL21 DE3 (New England Biolabs). Cells were selected on LB agar containing 30 μg/mL of kanamycin (Sigma). Positive clones were confirmed by Sanger sequencing at the Plateforme d'Analyse Génomique (IBIS) de l'Université Laval (Quebec City, QC, Canada). Overexpression was achieved by adding 0.4 mM IPTG to a mid-exponential culture containing 30 μg/mL of kanamycin. The incubation was at 25 °C with shaking (90 rpm) for 24 h.

## Purification of PaPBP3, SHV-1, and OXA-98

The bacterial lysis protocol followed that described above for *P. aeruginosa*, with some modifications. Briefly, after the 24 h induction with IPTG, the cultures were centrifuged at 3500 × *g*, 15 min, and lysed by French press after the lysozyme, DNase, RNase and anti-protease treatments. The pellets of SHV-1 and OXA-98 were resuspended in buffer A (300 mM NaCl, 25 mM Tris-HCl, pH 8 and 10 mM imidazole) and of PaPBP3 in Buffer B (400 mM NaCl, 25 mM Tris-HCl, pH 8, 10 mM imidazole, and 10% glycerol). Debris were removed by centrifugation at 10,000 × *g* and membrane proteins were separated from cytoplasm proteins by ultracentrifugation at 100,000 × *g* for 45 min. The supernatants containing SHV-1, OXA-98, and the soluble PaPBP3 were collected. Ni-NTA agarose column (Qiagen) were equilibrated with Buffer A or Buffer B before loading the supernatant. The column was washed with the respective buffers containing 40 mM imidazole. The protein was then eluted using buffer A or B containing 200 mM imidazole.

The eluted fractions were then concentrated using an Amicon® ultracentrifugal filter unit (MilliporeSigma) with a 10 kDa cutoff, and activity was assessed with 100 μM of nitrocefin for SHV-1 and OXA-98. Protein purity was verified by SDS-PAGE (see Fig. S2). Pure fractions were dialyzed in KP buffer (0.05 M, pH 7), with 20 mM NaHCO$_3$ added for the OXA-98[54]. Protein concentration was determined using the Bradford protein assay reagent (Bio-rad), and samples were aliquoted and stored at −80 °C.

## Kinetic constants of SHV-1, AmpC, and OXA-98

Purified β-lactamases were used to determine the $K_m$ and $K_{cat}$ constants in 0.05 M KP buffer (pH 7), with 20 mM NaHCO$_3$ added for OXA-98. Enzyme concentrations were 20 nM for AmpC and SHV-1, and 50 nM for OXA-98. Quartz cuvette (FireflySci Inc., Northport, NY) with 1 cm length path and 2 mm inside width was used with an Ultrospec 2100 Pro (Biochrom, Harvard Bioscience Inc., USA) to acquire the kinetic measures. Molar extinction coefficients were first determined after complete lysis of test antibiotics with SHV-1, using wavelengths that showed the highest change with the hydrolysis product. The following extinction coefficients (Δε) and wavelengths (λ) were used:

AMP Δε = 851 M$^{-1}$ cm$^{-1}$, λ = 235 nm; CEF Δε = 5225 M$^{-1}$ cm$^{-1}$, λ = 267 nm; LOR Δε = 11,182 M$^{-1}$ cm$^{-1}$, λ = 262 nm; BLOR Δε = 8980 M$^{-1}$ cm$^{-1}$, λ = 262 nm; MCEF Δε = 5560 M$^{-1}$ cm$^{-1}$, λ = 267 nm; and nitrocefin Δε = 17,400 M$^{-1}$ cm$^{-1}$, λ = 486 nm.

The steady-state parameters ($K_m$ and $K_{cat}$) were determined by plotting the initial velocity against substrate concentrations (Michaelis–Menten curve). The $K_{cat}$ was measured using the $V_{max}$ (μM/sec) divided by the total enzyme concentration and the catalytic efficiency was further calculated ($K_{cat}/K_m$). SHV-1 parameters were determined with three Michaelis–Menten curves of at least seven substrate concentrations, while OXA-98 and AmpC were duplicates of curves of at least seven substrate concentrations. The lysis of BAMP could not be detected, so its $K_m$ was measured indirectly as a $K_i$ using nitrocefin as a reporter. Initial velocities were measured at room temperature (25 °C). The $V_{max}$ and $K_m$ were

calculated using least squares fit regression in GraphPad version 10.2.2. $K_i$ was measured using a derivative of IC$_{50}$ with an adaptation of the Cheng–Prusoff equation[55]. To determine the $K_i$, BAMP and AMP were regarded as competitive inhibitors. At least five times the $K_m$ of nitrocefin was used to determine the initial velocities. The corrected $K_i$ was calculated using this formula:

$$K_i\,(corrected) = \frac{K_i\,(observed)}{1 + S/K_m NCF}$$

In this equation, the $K_i$ observed was obtained by plotting the reciprocal of initial velocity in μM/sec obtained with a fixed nitrocefin concentration (140 μM) and increasing concentrations of BAMP or AMP. The $K_i$ observed was corrected for the affinity of nitrocefin to the enzyme. In the equation, S is the nitrocefin concentration, and the $K_m$ of nitrocefin was as previously determined by a Michaelis–Menten curve.

## Antibiotic uptake assay

This method used ultra-performance liquid chromatography-mass spectrometry (UPLC-MS) to measure the amount of test antibiotics remaining in bacterial culture supernatants over time as a means to measure antibiotic uptake by bacterial cells. The method assumes that all antibiotic molecules in the medium (i.e., those that we measured by UPLC-MS) are not bound to the PBP targets in the periplasm, whereas it is assumed that the bactericidal effect (ΔCFU/mL) is correlated to antibiotic uptake and binding to PBPs. The test strain *E. coli* MC4100 was cultivated overnight in 10 mL of MHBCA or ID-MHBCA at 37 °C with shaking (200 rpm). The next day, the overnight culture was diluted in fresh preheated medium to obtain an OD$_{600}$ of ~0.7 in MHBCA or ID-MHBCA. The culture was further diluted 1:2 in MHBCA or ID-MHBCA (growth controls), or in MHBCA or ID-MHBCA containing the freshly prepared test antibiotic (2 × the final concentration) to obtain an initial OD$_{600}$ of ~0.35 (corresponding to 8.5 log10 CFU/mL) and 200 μM of antibiotic (final concentration). Antibiotics were also added to preheated MHBCA or ID-MHBCA without bacteria to control for the stability of the molecules in the medium over time. The moment bacteria were put in contact with the test antibiotic corresponded to the first time point (0 h). For each time point thereafter, 200 μL of sample from both assay and stability test tubes were retrieved and centrifuged at 4 °C for 5 min at 8000 × *g*. All supernatants were filtered (0.2 μm) and flash frozen in liquid nitrogen for transport. After melting, 2 μL of the samples were injected in the UPLC-MS before the analysis. In parallel, 20 μL of assay and growth control tubes were serially diluted in 96-well plates. Ten μL of each dilution was spotted on TSA plates and incubated for 24 h at 37 °C for CFU counting.

## Uptake assay data analysis

The concentration of the test antibiotic in the supernatant was measured with a standard curve prepared with the same drug preparation, both for MHBCA and ID-MHBCA. UPLC-MS analysis was conducted using a Waters UPLC-MS system (column Acquity UPLC® CSH™ C18 (2.1 × 50 mm) (Agilent Technologies, Santa Clara, CA, USA) packed with 1.7-μm particles) using acetonitrile (ACN) and water with 0.1% formic acid. The method used for 0 to 0.2 min was: 5% ACN; 0.2 to 1.5 min: 5% to 95% ACN; 1.5 to 1.8 min: 95% ACN; 1.8 to 2.0 min: 95 to 5% ACN; 2.0 to 2.5 min: 5% ACN in the ESI+ mode on a SQD2 Mass Spectrometer. The area under the curve of peaks corresponding to compounds of interest was integrated using UV integration in the MassLynx software (Agilent Technologies, Santa Clara, CA, USA). All data and statistical analyses were performed using GraphPad Prism version 10.2.2.

For analysis, the determined concentration in the supernatant was adjusted for the degradation of the molecules measured in the stability control tests for each time point. The final net antibiotic concentration in the medium was calculated at each time point by adding the concentration caused by degradation to the concentration of the test antibiotic measured in the assay tubes containing bacteria. The percentage of degradation of each antibiotic at each time points can be found in Table S7. A second control was

added for the β-lactam LOR and CEF to verify the impact of metabolic waste on the stability of the molecules. The supernatant from the 4-h growth control without antibiotic was collected by filtration 0.2 μm. LOR and CEF were then added to the filtrated supernatant at 200 μM, and degradation was measured after 3 h. The percentage of degradation for each condition are compiled in Table S7.

The bactericidal effect determined at each time point was the difference in the concentration of bacteria in the assay tube with antibiotic and that measured in the growth control tube without drug (ΔLog10 CFU/mL). The normal distribution of the data was verified with the Shapiro–Wilk test and the statistical analysis was a comparison of repeated measures ANOVA ($*p \leq 0.05$, $**p \leq 0.01$, $***p \leq 0.001$) between each time point relatively to the 0-h time point.

### Membrane PBPs and purified PaPBP3 binding assays

Membrane extracts containing PBPs were obtained as described above. Briefly, all species were grown in 1 L of TSB until the $OD_{600}$ reached 0.9. The washed pellet was incubated with lysozyme (400 μg/mL), DNase (10 μg/mL) RNase(10 μg/mL) and anti-protease cocktail (200 mg/20 mL). Lysis was done with a French press (20,000 psi) and membrane protein were pelleted by ultracentrifugation steps ($100,000 \times g$). The pellet was suspended in a minimal volume (300–500 μL) of KP 0.05 M buffer at pH 7. The protein concentration in the membrane preparations was determined with the Micro BCA TM Protein Assay Kit (Thermo Fisher Scientific, Rockford, IL) using bovine serum albumin as a standard.

The PBP binding assay was done as described previously[36,56,57]. For each antibiotic concentration tested, a quantity of 25 to 40 μg of membrane preparation was used as the source of PBPs, depending on the species. For the purified PaPBP3, 150 ng was used in the assay. First, increasing concentrations of the test antibiotic was added to the membrane or purified PaPBP3 preparations and incubated for 10 min at 30 °C. The reporter molecule Bocillin FL was then added at 12.5 μM for the membrane PBPs preparations and at 500 nM for the pure PaPBP3 samples for an additional 20 min at 30 °C. All reactions were stopped with the addition of the Laemmli-loading buffer (Bio-Rad) containing fresh β-mercaptoethanol (Sigma). All reactions were then heated at 100 °C for 3 min before migration on SDS-polyacrylamide discontinuous gel system (7.5% stacking and 10% separating). Control samples without Bocillin FL were also included to highlight autofluorescent proteins that are not PBPs.

### Antibiotic relative affinities ($IC_{50}$) for PBPs

To be able to visualize labeled PBPs after electrophoresis, gels were rinsed in water and fixed in a solution of 50% methanol and 7% acetic acid for 30 min. The fluorescent images of gels were obtained with the Typhoon FLA 9500 instrument (GE-Healthcare Bio-Sciences Inc., Baie-d'Urfé, QC, CA) using a 473 nm excitation wavelength. To obtain quantifiable data, the software Quantity One 1D analysis was used (version 4.6.6; Bio-Rad Laboratories, Richmond, CA). The volume (intensity × constant area) of each PBP was measured. The apparent $IC_{50}$ for each membrane PBP, and similarly for the $IC_{50}$ for the purified PaPBP3, was defined as the antibiotic concentration (μM) needed to reduce by 50% the binding of Bocillin FL. All $IC_{50}$ data were obtained from at least two independent PBP binding assays using two different sets of compound concentrations using a linear regression. The PBP profiles were confirmed, and the PBP numbering were determined using published data and individual molecular weights. Control antibiotics were also used to confirm PBP identification. For example, amdinocillin (Sigma) was used for PBP2 identification. In membrane fractions, the PBP2 of *P. aeruginosa* comigrated with a non-specific fluorescent protein. The IC50 for PBP 2 of *P. aeruginosa* was measured by including the SDS-PAGE comigrating autofluorescent protein in the background, as done previously[57].

### Statistics and reproducibility

All statistical analyses were performed using GraphPad Prism version 10.2.2. Normality was confirmed before applying for the ANOVA test. Data were presented as means ± standard deviation (SD). Statistical significance was expressed as $p$ values: $*p < 0.05$; $**p < 0.01$; $***p < 0.001$; ns not significant. Dunnett's multiple comparison tests were conducted to determine statistical significance between the groups specified in each section. For the uptake assay, repeated measures ANOVA was used to account for data dependencies over time. For this assay, a minimum of three biological replicates were done with a fresh antibiotic stock and fresh media (MHBCA and ID-MHBCA).

### Reporting summary

Further information on research design is available in the Nature Portfolio Reporting Summary linked to this article.

### Data availability

All data analysed during this study are included in this published article, in the Supplementary Information and Supplementary Data 1. The uncropped and unedited gels are included in Fig. S3. All raw data generated are available from the corresponding authors and plasmid generated in this study could be send on request.

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

## Acknowledgements

This work was supported by a Team Grant from the Fonds Québécois de la Recherche sur la Nature et Technologies (FRQNT) to FM (2019-PR-255463). We also thank a funding contribution from the Natural Sciences and Engineering Research Council of Canada (NSERC, discovery grant no. 2020-04811 to F.M.). E.L. received an Alexandre-Graham-Bell doctoral research studentship from NSERC (BESD3-556674-2020) and a grant from the Fonds Québécois de la Recherche sur la Nature et Technologies (FRQNT) (B2X – 334786). The sponsors had no role in the design, execution, interpretation, or writing of the study. We are most thankful to Hsiri Therapeutics for providing the synthetic siderophores used in this study and to Antoine Désilets for helping with the β-lactamase kinetic analysis.

## Author contributions

E.L., R.B., and H.G. performed the experiments. Y.-M.L., M.G., R.L., and M.J.M. synthesized and provided the characterization of the conjugates. F.M. and P.-L.B. revised and contributed to the analysis of the results. E.L. and R.B. wrote the manuscript, and F.M., M.J.M., and P.-L.B. contributed in the revision of it.

## Competing interests

The authors declare no competing interests. L.R., Y.-M.L., M.G., and M.J.M. are associated with Hsiri Therapeutics, which provided the synthetic siderophores used in this study. No funds were obtained from Hsiri.
