## [Transparent Peer Review file · Communications Biology]

Unraveling the Mechanisms Behind the Enhanced Efficacy of β -Lactam-based Sideromycins

Corresponding Author: Professor François Malouin

Version 0:

Reviewer comments:

Reviewer #1

(Remarks to the Author)

This manuscript addresses a significant biological and pharmacological question: how conjugating β -lactam antibiotics to siderophores can enhance their efficacy against Gram-negative bacteria. This is highly relevant given the escalating threat of antimicrobial resistance and the urgent need for new therapeutic approaches, particularly against WHO priority 1 pathogens.

The study investigates several key determinants of siderophore- β -lactam (SID- β L) conjugate activity: β -lactamase sensitivity, outer membrane permeability, efflux susceptibility, and affinity for penicillin-binding proteins (PBPs). The authors adopt a comparative approach, analyzing three SID- β L conjugates across four clinically relevant Gram-negative species—*Escherichia coli*, *Klebsiella pneumoniae*, *Pseudomonas aeruginosa*, and *Acinetobacter baumannii*.

The biological questions addressed are both timely and fundamental:

- How does siderophore-mediated uptake influence antibiotic entry into bacterial cells?
- Does conjugation enhance interaction with the antibiotic's target?
- Can it circumvent resistance mechanisms such as low permeability or efflux?

These questions are not only of biological interest but are also critical for the rational design of next-generation antibiotics. The manuscript clearly explores a timely and under-investigated area. Understanding how siderophore conjugation alters antibiotic uptake, target engagement, and susceptibility to resistance mechanisms is essential for future therapeutic development.

While I am not an expert in all the techniques employed, the authors appear to present a thorough and multifaceted analysis of the key parameters influencing antibiotic efficacy. The use of multiple bacterial species with different resistance profiles adds substantial depth to the findings.

The study shows that increased affinity for high-molecular-weight PBPs is a major determinant of enhanced activity against *E. coli* and *K. pneumoniae*. Conversely, in *P. aeruginosa* and *A. baumannii*, improved uptake via TonB-dependent transporters appears to be the critical factor, highlighting the importance of species-specific mechanisms. The data also indicate that conjugation does not protect β -lactams from hydrolysis by β -lactamases, although positional effects (e.g., at the C3 position of cephalosporins) may offer partial protection.

In conclusion, this is a well-executed and insightful study that sheds light on the mechanism of action of siderophore-antibiotic conjugates. It provides a valuable framework for the rational design of such conjugates and will be of broad interest to researchers in microbiology, pharmacology, and antimicrobial development.

Minor comments:

1. Stability of conjugates: Please clarify whether the stability of the conjugates was evaluated under periplasmic or extracellular conditions.

2. Iron depletion controls: Regarding the use of Chelex-treated medium, do you have evidence that the medium is truly iron-depleted? What control was used to verify this? Was the residual iron concentration measured after Chelex treatment? Has the Chelex resin been regenerated after use? If not, it may not effectively remove iron upon reuse.

3. Iron concentration units: Throughout the manuscript, iron concentration is expressed inconsistently—sometimes in molarity, sometimes in $\mu\text{g/mL}$. Please standardize to molarity, which is more scientifically informative.

4. Line 53: I suggest using the well-established abbreviation TBDT (TonB-dependent transporter) instead of LGP, which is rarely used in the literature.

5. Line 54: Rephrase to indicate that TonB-dependent transport requires the entire TonB-ExbB-ExbD complex, not just TonB alone.

6. Line 56: It would be more accurate to state that TBDTs differ in their binding specificity or selectivity rather than in their affinity, as their affinities for ferri-siderophore complexes generally fall within the nanomolar range.

7. Line 66: Please clarify whether the estimated value of $\sim 10^{-18}$ M refers to iron solubility or concentration in aqueous environments.

8. Table 1:

o Do you have an estimate of the iron concentration in MHBCA medium? Including this would be helpful.

o Clarify the meaning of (+) and (-): do they correspond to iron-supplemented and iron-depleted conditions, respectively? This should be explained in the legend.

9. Figure legends: Please specify in the figure legends that "ID" stands for iron-depleted.

Reviewer #2

(Remarks to the Author)

This study provides a comprehensive and systematic investigation into the mechanisms underlying the enhanced antibacterial activity of β -lactam-based sideromycins (SID- β L conjugates) against Gram-negative pathogens. The work highlights the critical role of improved PBP affinity in Enterobacteriaceae and TonB-dependent ligand-gated porin (LGP)-mediated uptake in *Pseudomonas aeruginosa* and *Acinetobacter baumannii*. The findings are scientifically robust, methodologically sound, and hold significant translational potential for combating antibiotic resistance. The manuscript performs well in terms of format and detail, but there are still some areas for improvement.

Major concerns:

1. The rationale for selecting bis-catechol vs. hydroxamate siderophores is not fully elucidated. How do structural differences influence LGP selectivity or PBP binding?
2. Expand beyond MHBCA/ID-MHBCA to include intermediate iron levels (0.01–1 $\mu\text{g/mL Fe}^{3+}$) to map dose-dependent LGP expression and conjugate uptake.
3. Include efflux pump inhibitors (e.g., PA β N for RND pumps) to confirm minimal efflux impact.
4. The Eagle effect observed with MCEF against *E. coli* is briefly mentioned but not fully explained. A deeper discussion of this phenomenon and its implications for the design of effective conjugates would be beneficial.

Minor

1. Label key functional groups (e.g., β -lactam ring, iron-chelating catechol/hydroxamate) in Figure 1 to aid reader comprehension.
2. In line 641, "Zn $^{2+}$ (ZNSO $_4$)" should be "Zn $^{2+}$ (ZnSO $_4$)";
3. In line 672, "phase (OD \sim 0.9)" should be "phase (OD $_{600}$ \sim 0.9)".

Version 1:

Reviewer comments:

Reviewer #1

(Remarks to the Author)

Thank you for the revised version of the manuscript. I am fully satisfied with the way the issues I had raised have been addressed.

Regarding the data involving the efflux pump inhibitor (PA β N), I agree with the authors that the results are highly variable and sometimes even contradictory depending on the bacterial strain. As such, they do not allow for any firm conclusions to be drawn. This variability is not surprising given that the precise mode of action of PA β N remains unclear. In addition to acting on efflux pumps, PA β N may have off-target effects, including potential interference with the TonB-dependent uptake machinery. Moreover, at higher concentrations, it has been shown to increase outer membrane permeability, further complicating the interpretation of its effects.

Given this uncertainty, and the fact that the authors also acknowledge the inconclusive nature of these results, I believe the best approach would be to leave these data in the supplementary material. They could be briefly described in the Results section, while clearly stating that no conclusion can be drawn due to the multiple possible mechanisms involved. In my opinion, it would be preferable not to discuss these findings further in the Discussion section, as they may lead to overinterpretation or confusion.

This approach would allow the authors to share their experimental observations transparently, without attributing undue weight to inconclusive findings.

Reviewer #2

(Remarks to the Author)

The authors have addressed all the reviewers' comments thoroughly and appropriately. The revisions, including additional data and clarifications, significantly enhance the manuscript's clarity, robustness, and scientific rigor. The key conclusions regarding species-specific mechanisms are well-supported by the data. No major flaws or unresolved issues remain. This revised manuscript meets the standards of Communications Biology and is recommended for publication.

ANSWERS TO REVIEWER'S COMMENTS:

Lacasse E, *et al.*

We sincerely thank you for your thoughtful and constructive comments on our manuscript, “Unraveling the Mechanisms Behind the Enhanced Efficacy of β -Lactam-based Sideromycins”. We have carefully considered all feedback and revised the manuscript accordingly. Below, we provide a detailed, point-by-point response to each comment, outlining the changes made or the rationale where changes were not made.

Reviewer #1 (Remarks to the Author):

This manuscript addresses a significant biological and pharmacological question: how conjugating β -lactam antibiotics to siderophores can enhance their efficacy against Gram-negative bacteria. This is highly relevant given the escalating threat of antimicrobial resistance and the urgent need for new therapeutic approaches, particularly against WHO priority 1 pathogens.

The study investigates several key determinants of siderophore- β -lactam (SID- β L) conjugate activity: β -lactamase sensitivity, outer membrane permeability, efflux susceptibility, and affinity for penicillin-binding proteins (PBPs). The authors adopt a comparative approach, analyzing three SID- β L conjugates across four clinically relevant Gram-negative species—*Escherichia coli*, *Klebsiella pneumoniae*, *Pseudomonas aeruginosa*, and *Acinetobacter baumannii*.

The biological questions addressed are both timely and fundamental:

- How does siderophore-mediated uptake influence antibiotic entry into bacterial cells?
- Does conjugation enhance interaction with the antibiotic's target?
- Can it circumvent resistance mechanisms such as low permeability or efflux?

These questions are not only of biological interest but are also critical for the rational design of next-generation antibiotics. The manuscript clearly explores a timely and under-investigated area. Understanding how siderophore conjugation alters antibiotic uptake, target engagement, and susceptibility to resistance mechanisms is essential for future therapeutic development.

While I am not an expert in all the techniques employed, the authors appear to present a thorough and multifaceted analysis of the key parameters influencing antibiotic efficacy. The use of multiple bacterial species with different resistance profiles adds substantial depth to the findings.

The study shows that increased affinity for high-molecular-weight PBPs is a major determinant of enhanced activity against *E. coli* and *K. pneumoniae*. Conversely, in *P. aeruginosa* and *A. baumannii*, improved uptake via TonB-dependent transporters appears to

be the critical factor, highlighting the importance of species-specific mechanisms. The data also indicate that conjugation does not protect β -lactams from hydrolysis by β -lactamases, although positional effects (e.g., at the C3 position of cephalosporins) may offer partial protection.

In conclusion, this is a well-executed and insightful study that sheds light on the mechanism of action of siderophore–antibiotic conjugates. It provides a valuable framework for the rational design of such conjugates and will be of broad interest to researchers in microbiology, pharmacology, and antimicrobial development.

Minor comments:

1. Stability of conjugates: Please clarify whether the stability of conjugates was evaluated under periplasmic or extracellular conditions.

ANSWER: The stability of conjugated and unconjugated antibiotics was evaluated during the uptake assay to ensure that the measured decrease in concentrations was accurate. Stability was assessed in two different ways: (1) in medium without bacteria, and (2) in filtered bacterial supernatant collected at the 4-hour time point of the assay. This approach allowed us to verify the stability of the antibiotics under extracellular conditions, as well as in the presence of enzymes from the periplasm and cytoplasm that may have been released due to bacterial lysis. See the paragraph: Metabolic byproducts had limited impact on the concentrations measured during the uptake assay (line 425) of the revised manuscript.

2. Iron depletion controls: Regarding the use of Chelex-treated medium, do you have evidence that the medium is truly iron-depleted? What control was used to verify this? Was the residual iron concentration measured after Chelex treatment? Has the Chelex resin been regenerated after use? If not, it may not effectively remove iron upon reuse.

Regarding the use of Chelex-treated medium, do you have evidence that the medium is truly iron-depleted? What control was used to verify this?

ANSWER: The supporting evidence and controls are presented in Figure S1, which shows that bacterial growth is restored in a dose-dependent manner with increasing iron concentrations in iron-depleted medium. Following the recommendation of the second reviewer, we also included the new Table S3 illustrating the effect of increasing iron concentrations on the MICs of the different bacterial species. As expected, higher iron concentrations led to a dose-dependent increase in MICs.

Was the residual iron concentration measured after Chelex treatment?

ANSWER: No, the iron concentration was not directly measured after Chelex treatment, the medium was prepared according to CLSI (Clinical and Laboratory Standards Institute) guidelines for ID-MHBCA. This standardized method is approved for Cefiderocol testing and

ensures an iron concentration of $\leq 0.03 \mu\text{g/mL}$ ($5.4 \times 10^{-7} \text{ M}$) Fe^{3+} .¹ The iron concentration in MHBCA has been estimated to be around $0.2 \mu\text{g/mL}$ ($3.6 \times 10^{-6} \text{ M}$).^{2,3}

Has the Chelex resin been regenerated after use? If not, it may not effectively remove iron upon reuse.

ANSWER: The resin used for Chelex treatment was always fresh and had never been regenerated, ensuring consistent iron removal. (This sentence has been added in the material and methods section- line 674 of the revised manuscript)

3. Iron concentration units: Throughout the manuscript, iron concentration is expressed inconsistently—sometimes in molarity, sometimes in $\mu\text{g/mL}$. Please standardize molarity, which is more scientifically informative.

ANSWER: Thank you for the comment. We agree that expressing concentration in molarity is scientifically more informative. However, we used the $\mu\text{g/mL}$ unit to facilitate comparison with other studies on Cefiderocol and Chelex-based iron depletion, where this unit is commonly reported. We have revised the text to include the equivalent molarity values but will retain the $\mu\text{g/mL}$ unit in the graph for consistency with the literature.

4. Line 53: I suggest using the well-established abbreviation TBDT (TonB-dependent transporter) instead of LGP, which is rarely used in literature.

ANSWER: The text has been modified (first example line 21)

5. Line 54: Rephrase to indicate that TonB-dependent transport requires the entire TonB-ExbB-ExbD complex, not just TonB alone.

ANSWER: The text in the introduction has been modified as follow (line 54 of the revised manuscript) :

TBDT require a TonB-ExbB-ExbD protein complex and the proton motive force to transport siderophores into the periplasm.

6. Line 56: It would be more accurate to state that TBDTs differ in their binding specificity or selectivity rather than in their affinity, as their affinities for ferri-siderophore complexes generally fall within the nanomolar range.

ANSWER: We agree, the text in the introduction has been modified (in **bold** below) (line 56 of the revised manuscript):

TBDTs are redundant across species and differ in their **specificity** for siderophore ligands

7. Line 66: Please clarify whether the estimated value of $\sim 10^{-18} \text{ M}$ refers to iron solubility or concentration in aqueous environments.

ANSWER: The sentence in the introduction has been modified (line 67 of the revised manuscript):

In aqueous solution and at neutral pH, iron solubility is low, and **ferric iron concentration** is estimated to be around 10^{-18} M.

8. Table 1:

Do you have an estimate of the iron concentration in MHBCA medium? Including this would be helpful.

ANSWER: Yes, the iron concentration in MHBCA is estimated to be around $0.2 \mu\text{g/mL}$ (3.6×10^{-6} M). This estimation also correlates with the MIC values measured at increasing iron concentrations in ID-MHBCA (see Table S3). The addition of $0.1 \mu\text{g/mL}$ of iron (1.8×10^{-6} M) in ID-MHBCA resulted in MICs equivalent to those observed with regular MHBCA. The text in the result section has been modified (in **bold**) to include these data (line 141 of the revised manuscript):

For each bacterial species, the MIC of at least one of the tested β -lactams was greatly improved when conjugated to a siderophore moiety, as revealed by the fold-improvement ratios from 8 to >8192 reported for wild-type strains in Table 1. It was expected that a lower iron concentration would increase the activity of the SID- β L conjugates due to a higher expression of TBDTs and enhanced outer membrane uptake. However, Table 1 shows only two cases where a low iron concentration enhanced the SID- β L activity by more than 4-fold when compared to the non depleted medium. In the first case, BAMP showed a lower MIC against *P. aeruginosa*, in ID-MHBCA (MIC of $0.1 \mu\text{M}$) compared to that measured in MHBCA (MIC of $1.6 \mu\text{M}$). In the second case, the low iron concentration also improved the activity of MCEF in ID-MHBCA (MIC $0.2 \mu\text{M}$) compared to that seen in MHBCA ($1.6 \mu\text{M}$) against *A. baumannii*. **However, a dose-dependent increase in the MICs of the conjugates was observed in all species following iron supplementation of chelex treated ID-MHBCA medium (0, 0.1, and 1 $\mu\text{g/mL}$) (see Table S3). Supplementation with $0.1 \mu\text{g/mL}$ (1.8×10^{-6} M) resulted in MIC values equivalent to those measured in untreated MHBCA. This correlates with the previously measured iron concentration of $0.24 \mu\text{g/mL}$ (4.3×10^{-6} M) in this medium.² Moreover, supplementation with $1 \mu\text{g/mL}$ (1.8×10^{-5} M) led to a further increase in MICs, correlating with a decrease TBDTs expression compared to MHBCA.**

Clarify the meaning of (+) and (-): do they correspond to iron-supplemented and iron-depleted conditions, respectively? This should be explained in the legend.

ANSWER: The legends have been modified to clarify that “+” means MHBCA and “-” means iron-depleted-MHBCA. (first example line 169).

9. Figure legends: Please specify in the figure legends that “ID” stands for iron-depleted.

ANSWER: The legends have been modified (first example line 169).

Reviewer #2 (Remarks to the Author):

This study provides a comprehensive and systematic investigation into the mechanisms underlying the enhanced antibacterial activity of β -lactam-based sideromycins (SID- β L conjugates) against Gram-negative pathogens. The work highlights the critical role of improved PBP affinity in Enterobacteriaceae and TonB-dependent ligand-gated porin (LGP)-mediated uptake in *Pseudomonas aeruginosa* and *Acinetobacter baumannii*. The findings are scientifically robust, methodologically sound, and hold significant translational potential for combating antibiotic resistance. The manuscript performs well in terms of format and detail, but there are still some areas for improvement.

Major concerns:

1. The rationale for selecting bis-catechol vs. hydroxamate siderophores is not fully elucidated. How do structural differences influence LGP selectivity or PBP binding?

ANSWER: Different types of siderophore have been conjugated to β -lactams in the past.⁴ The bis-catechol used in this study showed broad spectrum activity against a wide range of Gram-negative pathogens. Moreover, it showed enhanced activity compared to the unconjugated β -lactam even if the bacteria were naturally resistant. This phenomenon was not seen with hydroxamate type siderophores. Moreover, the activity of conjugated β -lactams to synthetic hydroxamate-siderophore never surpass the activity of the unconjugated one⁵.

Considering this information the text will be modified in the introduction (**in bold** below) to justify the siderophore choices (line 98 of the revised manuscript):

To investigate the mechanisms underlying the enhanced potency conferred by adding a siderophore to an antibiotic, this study examined three siderophore- β -lactam (SID- β L) conjugates: a bis-catechol siderophore, azotochelin⁶, conjugated to ampicillin (BAMP)⁷ or conjugated to loracarbef (BLOR), and a mixed ligand bis-catechol-mono-hydroxamate siderophore⁸ conjugated to cefaclor (MCEF); see Fig. 1d-1f. The design of the mixed siderophore used in MCEF was inspired by fimsbactin (Fig. 1c), a siderophore synthesized by *Acinetobacter baumannii*⁸. **The synthetic siderophores used in this study were the result of many years of optimization.** Previous studies have demonstrated that their conjugation to β -lactam, increased the antibiotic potency, against clinically relevant bacterial species, by over 500-fold compared to the unconjugated one⁹. **In contrast, other siderophores, such as those based on**

hydroxamates, exhibit only low to moderate activity when conjugated to β -lactams, and the potency of these conjugates does not surpass that of the unconjugated β -lactam ⁵.

2. Expand beyond MHBCA/ID-MHBCA to include intermediate iron levels (0.01–1 $\mu\text{g/mL}$ Fe^{3+}) to map dose-dependent LGP expression and conjugate uptake.

ANSWER: A table (Table S3) has been added to the supplementary information section. Iron supplementation led to a dose-dependent increase of MIC for conjugated β -lactams and had no impact on unconjugated ones. The text has been modified (**in bold**) to incorporate the new data (line 141 of the revised manuscript):

For each bacterial species, the MIC of at least one of the tested β -lactams was greatly improved when conjugated to a siderophore moiety, as revealed by the fold-improvement ratios from 8 to >8192 reported for wild-type strains in Table 1. It was expected that a lower iron concentration would increase the activity of the SID- β L conjugates due to a higher expression of TBDTs and enhanced outer membrane uptake. However, Table 1 shows only two cases where a low iron concentration enhanced the SID- β L activity by more than 4-fold when compared to the non depleted medium. In the first case, BAMP showed a lower MIC against *P. aeruginosa*, in ID-MHBCA (MIC of 0.1 μM) compared to that measured in MHBCA (MIC of 1.6 μM). In the second case, the low iron concentration also improved the activity of MCEF in ID-MHBCA (MIC 0.2 μM) compared to that seen in MHBCA (1.6 μM) against *A. baumannii*. **However, a dose-dependent increase in the MICs of the conjugates was observed in all species following iron supplementation (0, 0.1, and 1 $\mu\text{g/mL}$) of chelex treated ID-MHBCA medium (Table S3). Supplementation with 0.1 $\mu\text{g/mL}$ (1.8×10^{-6} M) resulted in MIC values equivalent to those measured in untreated MHBCA. This correlates with the previously measured iron concentration of 0.24 $\mu\text{g/mL}$ (4.3×10^{-6} M) in this medium.² Moreover, supplementation with 1 $\mu\text{g/mL}$ (1.8×10^{-5} M) led to a further increase in MICs, correlating with a decrease TBDTs expression compared to MHBCA.**

3. Include efflux pump inhibitors (e.g., PA β N for RND pumps) to confirm minimal efflux impact.

ANSWER: We did try the PA β N with *E. coli*, *A. baumannii* and *P. aeruginosa*. PA β N had no effect on the MICs of any unconjugated β -lactams tested in all species. Unexpectedly, the results were more difficult to explain for conjugates and will probably necessitate more assays to explain them. In fact, combining PA β N with the conjugated β -lactam in *A. baumannii* and *E. coli*, antagonized their activity (between 8 and ≥ 128 -fold). This might be because it counteracts the active transport mechanism in those strains and might influence the TonB complex, but we cannot prove this hypothesis in this study. In *P. aeruginosa*, it improved the

activity of BAMP and its MICs decreased by 8-fold but it has no effect on AMP and all the other antibiotics. We could add these data in supplementary information (new Table S5) but won't be able to explain them completely. The text could be modified as follows:

Result section (line 315 of the revised manuscript):

Efflux was investigated as a potential factor influencing the efficacy of conjugated β -lactams, either by enhancing their effectiveness or contributing to their lack of activity. To quantify the impact of efflux pumps on the activity of the antibiotics, MICs were tested in combination with the non-specific efflux pumps inhibitor phenylalanine-arginine- β -naphthylamide (PA β N). PA β N has been optimized to counteract *P. aeruginosa* efflux pumps. In this species, the addition of 25 μ g/mL of PA β N potentiated the activity of BAMP by 8-fold. However, it antagonized the activity of all conjugates in other species, leading to an inhibition of activity against *A. baumannii* (see results in Table S5). On the other hand, the bulky control antibiotics used in this assay (rifampicin and erythromycin) saw their activity improved in the presence of PA β N, possibly due to a combination of efflux pumps inhibition and PA β N membrane permeabilization, as previously suggested.¹⁰ As such, a possible explanation to the loss of activity of conjugates in presence of PA β N (opposite effect to that observed for rifampicin and erythromycin) might be a secondary target affecting the active transport of siderophores through TBDTs. However, further tests are needed to confirm this hypothesis. To complement the PA β N assays, a variety of efflux pump mutants of *E. coli* and *P. aeruginosa* (see Table S6 for details) were analyzed.

Discussion section (line 622 of the revised manuscript):

The results obtained to evaluate the importance of resistance factors such as β -lactamases and efflux pumps were useful in explaining intrinsic resistance but did not account for the enhanced efficacy of conjugated β -lactams compared to their unconjugated forms. For example, the resistance of *K. pneumoniae* to AMP and MCEF could be attributed to the catalytic efficiency (K_{cat}/K_m) of its β -lactamases, as well as the increased activity observed when these antibiotics were combined with clavulanic acid. Also, BAMP activity increased when combined with PA β N in MHBCA, although it was already active. However, PA β N antagonized all conjugates in other species, likely due to off-target effects, as previously reported in *E. coli*¹¹. This could lead to the leakage of AmpC β -lactamase or interfere with the active transport of siderophores and conjugates, but more tests are needed to confirm those hypotheses. Nevertheless, these findings should be considered for future combination therapies involving an efflux pump inhibitor and a siderophore- β -lactam conjugate. Additionally, further studies investigating a broader range of efflux pumps may be necessary. For instance, the

MuxABC-OpmB system in *P. aeruginosa* has been shown to interfere with the activity of cefiderocol by promoting the excretion of the endogenous siderophore pyoverdine.

4. The Eagle effect observed with MCEF against *E. coli* is briefly mentioned but not fully explained. A deeper discussion of this phenomenon and its implications for the design of effective conjugates would be beneficial.

ANSWER: The manuscript was modified as follows (in **bold** below) in the result section and the discussion:

Result section (line 163 of the revised manuscript):

Nevertheless, MCEF was still partly active against *E. coli* in MHBCA (growth inhibition in the concentration range indicated in brackets in Table 1) although growth was observed at higher concentrations (>25 μM). It is suspected that this phenomenon is the already documented Eagle effect. **In *Proteus vulgaris*, this effect as been attributed to the induction of β -lactamase expression.¹² Whereas, in *E. coli*, this phenomenon has been attributed mainly to the SOS response and the upregulation of genes implicated in enterobactin biosynthesis.¹³** In ID-MHBCA, MCEF completely lost its activity against *E. coli*. The Eagle effect and this inactivity in ID-MHBCA was further investigated by the study of relevant mutants in the next section.

Discussion section (line 583 of the revised manuscript):

For MCEF, even if an increased binding to PBP3 was measured, the limited uptake in MHBCA and the Eagle effect mostly caused by enterobactin expression (Table 2) in ID-MHBCA, prevented its activity. **A study in *P. vulgaris* measuring the impact of the Eagle effect seen *in vitro* showed that high β -lactamase induction also happened *in vivo* leading to a reduce survival of infected mice at higher doses of cefmenoxime.¹⁴ This could be of importance for siderophore- β -lactam conjugates inducing this phenomenon as a small dose might be more efficient than high doses to treat infections.**

Minor

1. Label key functional groups (e.g., β -lactam ring, iron-chelating catechol/hydroxamate) in Figure 1 to aid reader comprehension.

ANSWER: With the siderophore and the antibiotic highlighted in different colors and the numbered atoms, we think the figure is already quite detailed, and adding additional elements might make it visually overwhelming. However, we have slightly improved the legend for clarity.

2. In line 641, “Zn²⁺(ZNSO₄)” should be “Zn²⁺(ZnSO₄)”;

ANSWER: The text has been modified.

3. In line 672, “phase (OD ~ 0.9)” should be “phase (OD600 ~ 0.9)”.

ANSWER: The text has been modified.

Table S3. Minimal inhibitory concentrations (MICs) of conjugated and unconjugated β -lactams in Chelex-treated MHBCA (ID-MHBCA) supplemented with increasing iron concentrations

MICs ^a of conjugated and unconjugated β -lactam in MHBCA and in ID-MHBCA with increasing supplemental iron concentrations							
	ID-MHBCA iron supplement in $\mu\text{g/mL}$ and MIC ratio	AMP	BAMP	LOR	BLOR	CEF	MCEF
E. coli ATCC 25922	0	13	0.05 ^b	1.6	0.012	3.1	>200
	0.1	ND	0.1	3.1	0.05	3.1	ND
	1	25	0.2	3.1	0.1	ND	25
	MHBCA	13	0.1	3.1	0.024	3.1	[6.3-25]
	MIC ratio 1/0	2	4	2	8	1	ND
K. pneumoniae ATCC 13883	0	>200	>200	1.6	0.1	3.1	>200
	0.1	ND	ND	1.6	0.4	3.1	ND
	1	>200	>200	3.1	6.3	6.3	>200
	MHBCA	>200	>200	1.6	0.2	3.1	>200
	MIC ratio 1/0	ND	ND	2	64	2	ND
P. aeruginosa ATCC 27853	0	>200	0.1	>200	>200	>200	>200
	0.1	ND	0.8	ND	>200	ND	ND
	1	>200	13	>200	>200	>200	>200
	MHBCA	>200	1.6	>200	13	>200	>200
	MIC ratio 1/0	ND	128	ND	ND	ND	ND
A. baumannii ATCC 19606	0	>200	0.2	>200	0.024	>200	0.2
	0.1	ND	0.4	ND	0.1	ND	1.6
	1	>200	1.6	>200	0.4	>200	>6.3
	MHBCA	>200	0.4	>200	0.05	>200	1.6
	MIC ratio 1/0	ND	8	ND	16	ND	>32

^a MICs are given in μM , Chelex-treated iron-deprived-MHBCA; ID-MHBCA, AMP; ampicillin, BAMP; bis-catechol-ampicillin, CEF; cefaclor, MCEF; mixed bis-catechol-mono-hydroxamate-cefaclor, LOR; loracarbef, BLOR; bis-catechol-loracarbef, ND, not determined, ID-MHBCA: iron deprived-MHBCA.

^b In bold, are the ratios in which iron supplementation led to a dose-dependent increase in MIC leading to a ≥ 4 -fold difference in bacteria susceptibility when comparing tests performed in 0 and 1 $\mu\text{g/mL}$ supplemental iron.

Table S5. Impact of efflux pump inhibitor phenylalanine-arginine- β -naphthylamide (PA β N) on the activity of β -lactam-siderophore conjugates.

Antibiotics ^b	MICs ^a of conjugated and unconjugated β -lactam in combination with Pa β N in MHBCA					
	E. coli MC4100		P. aeruginosa ATCC 27853		A. baumannii ATCC 19606	
	- Pa β N ^c	+ PA β N	- PA β N	+ PA β N	- PA β N	+ PA β N
AMP	13	13	>200	>200	>200	>200
BAMP	0.024	0.2	1.6	0.2	0.4	>6.3
LOR	3.1	6.3	>200	>200	>200	>200
BLOR	0.024	0.2	13	>200	0.05	>6.3
CEF	6.3	13	>200	>200	>200	>200
Mix-CEF	6.3	>200	>200	>200	1.6	>6.3
ERY	128	16	>128	32	ND	ND
RIF	ND	ND	16	2	16	\leq 0.012

^a MICs are expressed in μ M except, ERY and RIF that are in μ g/mL.

^b AMP; ampicillin, BAMP; bis-catechol-ampicillin, LOR; loracarbef, BLOR; bis-catechol-loracarbef, CEF; cefaclor, MCEF; mixed bis-catechol-mono-hydroxamate-cefaclor, ERY; erythromycin, RIF; rifampicin, ND; not determined.

^c - Pa β N; without phenylalanine-arginine- β -naphthylamide, + PA β N; with 25 μ g/mL phenylalanine-arginine- β -naphthylamide.

References

1. Simner, P. J. & Patel, R. Cefiderocol Antimicrobial Susceptibility Testing Considerations: the Achilles' Heel of the Trojan Horse? *J Clin Microbiol* **59**, e00951-20 (2020).
2. Ito, A. *et al.* Siderophore cephalosporin cefiderocol utilizes ferric iron transporter systems for antibacterial activity against *Pseudomonas aeruginosa*. *Antimicrob Agents Chemother* **60**, 7396–7401 (2016).
3. Hackel, M. A. *et al.* Reproducibility of broth microdilution MICs for the novel siderophore cephalosporin, cefiderocol, determined using iron-depleted cation-adjusted Mueller-Hinton broth. *Diagn Microbiol Infect Dis* **94**, 321–325 (2019).
4. Lin, Y. M., Ghosh, M., Miller, P. A., Möllmann, U. & Miller, M. J. Synthetic sideromycins (skepticism and optimism): selective generation of either broad or narrow spectrum Gram-negative antibiotics. *BioMetals* **32**, 425–451 (2019).
5. Al Shaer, D., Al Musaimi, O., de la Torre, B. G. & Albericio, F. Hydroxamate siderophores: Natural occurrence, chemical synthesis, iron binding affinity and use as Trojan horses against pathogens. *Eur J Med Chem* **208**, 112791 (2020).
6. Cornish, A. S. & Page, W. J. The catecholate siderophores of *Azotobacter vinelandii*: their affinity for iron and role in oxygen stress management. *Microbiology (N Y)* **144**, 1747–1748.
7. Liu, R. *et al.* A synthetic dual drug sideromycin induces Gram-negative bacteria to commit suicide with a Gram-positive antibiotic. *J Med Chem* **61**, 3845–3854 (2018).
8. Wencewicz, T. A. & Miller, M. J. Biscatecholate-monohydroxamate mixed ligand siderophore-carbacephalosporin conjugates are selective sideromycin antibiotics that target *Acinetobacter baumannii*. *J Med Chem* **56**, 4044–4052 (2013).
9. Lin, Y. M., Ghosh, M., Miller, P. A., Möllmann, U. & Miller, M. J. Synthetic sideromycins (skepticism and optimism): selective generation of either broad or narrow spectrum Gram-negative antibiotics. *BioMetals* **32**, 425–451 (2019).
10. Lamers, R. P., Cavallari, J. F. & Burrows, L. L. The Efflux Inhibitor Phenylalanine-Arginine Beta-Naphthylamide (PAβN) Permeabilizes the Outer Membrane of Gram-Negative Bacteria. *PLoS One* **8**, (2013).

11. Matsumoto, Y. *et al.* Evaluation of Multidrug Efflux Pump Inhibitors by a New Method Using Microfluidic Channels. *PLoS One* **6**, e18547 (2011).
12. Ikeda, Y. & Nishino, T. Paradoxical Antibacterial Activities of β -Lactams against *Proteus vulgaris*: Mechanism of the Paradoxical Effect. *Antimicrob Agents Chemother* **32**, 1073–1077 (1988).
13. Lai, Y. H., Franke, R., Pinkert, L., Overwin, H. & Brönstrup, M. Molecular Signatures of the Eagle Effect Induced by the Artificial Siderophore Conjugate LP-600 in *E. coli*. *ACS Infect Dis* **9**, 567–581 (2023).
14. Ikeda, Y., Fukuoka, Y., Motomura, K., Yasuda, T. & Nishino, T. Paradoxical activity of β -lactam antibiotics against *Proteus vulgaris* in experimental infection in mice. *Antimicrob Agents Chemother* **34**, 94–97 (1990).

ANSWERS TO REVIEWER'S COMMENTS:

Lacasse E, *et al.*

We sincerely thank you for your thoughtful and constructive comments on our manuscript, “Unraveling the Mechanisms Behind the Enhanced Efficacy of β -Lactam-based Sideromycins”. Below, we provide a detailed response to the last comment of the reviewer.

REVIEWERS' COMMENTS:

Reviewer #1 (Remarks to the Author):

Thank you for the revised version of the manuscript. I am fully satisfied with the way the issues I had raised have been addressed

Regarding the data involving the efflux pump inhibitor (PA β N), I agree with the authors that the results are highly variable and sometimes even contradictory depending on the bacterial strain. As such, they do not allow for any firm conclusions to be drawn. This variability is not surprising given that the precise mode of action of PA β N remains unclear. In addition to acting on efflux pumps, PA β N may have off-target effects, including potential interference with the TonB-dependent uptake machinery. Moreover, at higher concentrations, it has been shown to increase outer membrane permeability, further complicating the interpretation of its effects.

Given this uncertainty, and the fact that the authors also acknowledge the inconclusive nature of these results, I believe the best approach would be to leave these data in the supplementary material. They could be briefly described in the Results section, while clearly stating that no conclusion can be drawn due to the multiple possible mechanisms involved. In my opinion, it would be preferable not to discuss these findings further in the Discussion section, as they may lead to overinterpretation or confusion.

This approach would allow the authors to share their experimental observations transparently, without attributing undue weight to inconclusive findings.

Answer: The text in the result section has been modified as follows (in bold) and the text in the discussion section has been removed:

Result section (line 296):

Efflux was investigated as a potential factor influencing the efficacy of conjugated β -lactams, either by enhancing their effectiveness or contributing to their lack of activity. To quantify the impact of efflux pumps on the activity of the antibiotics, MICs were tested in combination

with the non-specific efflux pumps inhibitor phenylalanine-arginine- β -naphthylamide (PA β N). PA β N has been optimized to counteract *P. aeruginosa* efflux pumps. In this species, the addition of 25 μ g/mL of PA β N potentiated the activity of BAMP by 8-fold. However, it antagonized the activity of all conjugates in other species, leading to an inhibition of activity against *A. baumannii* (see results in Table S4). On the other hand, the control antibiotics used in this assay (rifampicin and erythromycin) saw their activity improved in the presence of PA β N, possibly due to a combination of efflux pumps inhibition and PA β N membrane permeabilization, as previously suggested.³¹ As such, a possible explanation to the loss of activity of conjugates in presence of PA β N (opposite effect to that observed for rifampicin and erythromycin) might be a secondary target affecting the active transport of siderophores through TBDTs. **However, further tests are needed to confirm this hypothesis, and no conclusions could be drawn.** To complement the PA β N assays, a variety of efflux pump mutants of *E. coli* and *P. aeruginosa* (see Table S5 for details) were analyzed.

Discussion section: (the crossed text has been removed-line 558)

The results obtained to evaluate the importance of resistance factors like β -lactamases and efflux pumps were helpful to explain intrinsic resistance but did not account for the enhanced efficacy of conjugated β -lactams compared to their unconjugated forms. For example, the resistance of *K. pneumoniae* to AMP and MCEF could be attributed to the catalytic efficiency (K_{cat}/K_m) of its β -lactamases, as well as the observed activity increase when these antibiotics were used in combination with clavulanic acid. ~~The results obtained to evaluate the importance of resistance factors such as β -lactamases and efflux pumps were useful in explaining intrinsic resistance but did not account for the enhanced efficacy of conjugated β -lactams compared to their unconjugated forms. For example, the resistance of *K. pneumoniae* to AMP and MCEF could be attributed to the catalytic efficiency (K_{cat}/K_m) of its β -lactamases, as well as the increased activity observed when these antibiotics were combined with clavulanic acid. Also, BAMP activity increased when combined with PA β N in MHBCA, although it was already active. However, PA β N antagonized all conjugates in other species, likely due to off target effects, as previously reported in *E. coli*¹¹. This could lead to the leakage of AmpC β -lactamase or interfere with the active transport of siderophores and conjugates, but more tests are needed to confirm those hypotheses. Nevertheless, these findings should be considered for future combination therapies involving an efflux pump inhibitor and a siderophore β -lactam conjugate.~~ Additionally, further studies investigating a broader range of efflux pumps may be necessary. For instance, the MuxABC-OpmB system in *P. aeruginosa* has been shown to interfere with the activity of cefiderocol by promoting the excretion of the endogenous siderophore pyoverdine.